# Investigation of new particle formation at the summit of Mt. Tai, China

Ganglin Lv[1], Xiao Sui[1], Jianmin Chen[1,2*], Rohan Jayaratne[3], Abdelwahid Mellouki[1,4]

[1]School of Environmental Science and Engineering, Environment Research Institute, Shandong University, Jinan, Shandong 250100, China

[2]Shanghai Key Laboratory of Atmospheric Particle Pollution and Prevention (LAP3), Institute of Atmospheric Sciences, Fudan University, Shanghai 200433, China

[3]International Laboratory for Air Quality and Health, Science and Engineering Faculty, Queensland University of Technology, GPO Box 2434, Brisbane QLD 4001, Australia

[4]Institut de Combustion, Aérothermique, Réactivité et Environnement, CNRS, 45071 Orléans cedex 02, France

*Correspondence to*: Jianmin Chen (jmchen@fudan.edu.cn); Tel.: +86 53188363711; fax: +86 531 88361990

**Abstract.** To date few comprehensive field observations of new particle formation (NPF) have been carried out on the mountain-top sites in China. Simultaneous measurements of particle size distribution, gas species, meteorological parameters, mass concentration and chemical composition of $PM_{2.5}$ were performed at the summit of Mt. Tai (1534 m ASL) from 25 July to 24 August 2014 (I), 21 September to 9 December 2014 (II), and 16 June to 7 August 2015 (III), to investigate the characteristics and favorable factors of NPF in the relatively clean mountain-top environment. NPF events were identified based on the particle size distribution, and 66 such events were observed in a period of 164 days - corresponding to an occurrence frequency of 40 %. Formation rates of 3 nm particles ($J_3$) and growth rates (GR) were in the range of 0.82-25.04 $cm^{-3}$ $s^{-1}$ and 0.58-7.76 nm $h^{-1}$, respectively. On the average, the condensation sink (CS), $O_3$ concentration, air temperature and relative humidity were lower, whereas $SO_2$ concentration was higher on NPF days compared with non-NPF days. The onset of NPF events at the summit of Mt. Tai might not be limited by initial sulfuric acid concentration because there was no prominent superiority in sulfuric acid proxy concentration between NPF days and non-NPF days in the early morning. NPF events were more common in the east-southeast and west-southwest wind directions, which were mainly attributed to the relatively low CS in the east-southeast and elevated $SO_2$ concentration in the west-southwest. Back trajectory analysis indicated that the continental air mass dominated on both NPF days and non-NPF days. The transported air masses passing through the heavily polluted areas could enhance the occurrence of NPF, which was possibly associated with high level of anthropogenic species carried from the polluted regions. Four NPF events were observed during haze episodes, in the presence of the elevated $PM_{2.5}$ and trace gases concentrations simultaneously. A case study on 11 November 2014 was specially presented, where the NPF haze event was probably driven by the enhanced solar radiation at noon and changed new air mass bringing rich precursors.

**Keywords.** New particle formation; Mountain observations; Favorable conditions; Haze episodes

**1 Introduction**

Atmospheric aerosols play a critical role in affecting global radiation balance and climate, directly through scattering and absorption of solar radiation, and indirectly by modifying cloud properties as potential cloud condensation nuclei (CCN) (Kuang et al., 2010). Aerosol particles are involved in several atmospheric chemistry processes such as enhancing haze and decreasing visibility, and they can also harm human health by inhalation (Han, 2012). Previous studies have showed that the nucleation of atmospheric gas-phase precursors and the subsequent growth to larger particles, widely known as new particle formation (NPF), is the largest source of atmospheric aerosol particles (Zhang et al., 2012). Field observations have exhibited that NPF typically increase the particle number concentration by a factor between two and ten (Gong et al., 2010). Model studies also revealed that NPF accounted for 5-50 % of CCN in the lower boundary layer (Spracklen et al., 2008). An in-depth study of the process of NPF and its effects could help control atmospheric aerosol pollution in China.

With the development of instruments that measure particle size distribution, NPF events have been widely observed all over the world in recent decades. These observation sites include northern-most sub-arctic, remote boreal forests, industrialized agricultural regions, high-iodine coastal environments, and polluted urban areas (Dal Maso et al., 2002). The frequency of NPF events varies significantly between locations. Hallar et al. (2011) reported that NPF events in urban areas, such as Pittsburgh, occurred on about 35-50 % of all days, whereas the corresponding value for the remote background sites in Finland and Sweden was just 2-27 %. Manninen et al. (2010) found that the frequency of NPF events ranged from 21 % to 57 % at the twelve field sites around Europe, and the number of observed NPF days was closely related to the regional atmospheric conditions.

In the past decade, many campaigns and studies on NPF have been carried out in China. NPF events were first reported in China in 2004 by Wehner et al. (2004) who used a Twin Differential Mobility Particle Sizer (TDMPS). Soon after, Liu et al. (2008) made observations of NPF at a rural/coastal site in XinKen (Guangdong Province). In 2005, Gao et al. (2009) investigated the occurrence of NPF in a suburban environment in the Yangtze River delta using a Wide-range Particle Spectrometer (WPS). Thereafter, several observations of NPF have been reported in urban/suburban/rural environments around China (An et al., 2015; Wang et al., 2011). However, there have not been many observations of NPF on the mountain-top sites in China so far. Zhang et al. (2016) reported the occurrence of NPF on Mt. Huang (1840 m ASL) with a WPS instrument from April to July 2008. However, these current studies had significant limitation in measurement methods and seasonal variation. It is clear that multi-season observations using nanometer-scale instruments were essential and valuable for NPF research in mountain environments.

Particle formation and growth rates vary between different field environments. Kulmala et al. (2004), reviewing a number of studies, found that the typical formation rate was in the range of 0.01-10 $cm^{-3}$ $s^{-1}$. In urban areas it may be about 100 $cm^{-3}$ $s^{-1}$, while in coastal zones it can be as high as $10^4$-$10^5$ $cm^{-3}$ $s^{-1}$. Typical growth rate of newly formed particles ranges from 1

to 20 nm h$^{-1}$, and at some coastal areas it is as high as 200 nm h$^{-1}$.

Previous researches have shown that sulfuric acid, ammonia, organic vapors, and iodide species in the atmosphere were involved in the nucleation process under specific conditions. Gaseous sulfuric acid was the most critical candidate that participated in binary, ternary and ion induced nucleation (Boy et al., 2005; Wang et al., 2011; Zhang, 2010; Saunders et al., 2010; Allan et al., 2015). It has also been shown that the nucleation rate is a function of sulfuric acid concentration with a power dependency exponent, whose exponent varied significantly between different nucleation theories (Kulmala et al., 2006; Wang et al., 2011). However, the occurrence of NPF cannot be determined by a single sulfuric acid, and other factors such as pre-existing particles, trace gases, meteorological conditions, and air mass transport should also be considered. Comprehensive investigations of these factors affecting the occurrence of NPF in China have mainly been conducted in urban/suburban/rural environments. Since the mechanism of NPF under heavily polluted conditions is significantly different from that under relatively clean conditions (Kulmala et al., 2016; Hu et al., 2016; Wang et al., 2013), the comprehensive investigation of NPF on relatively clean mountain-top sites is of great important in China. In addition, favorable conditions for NPF events were characterized by different locations (Nie et al., 2014; Wang et al., 2014a; An et al., 2015), thus an intensive analysis for specific sites, such as Mt. Tai in this study, is also vital.

In this paper, it presents the results of the intensive field observations at the summit of Mt. Tai (1534 m ASL) which is a relatively clean mountain-top environment. This study was based on simultaneous measurements of particle size distribution, meteorological conditions, gaseous species, mass concentration and chemical composition of PM$_{2.5}$ during three campaigns (25 July to 24 August 2014, I; 21 September to 9 December 2014, II; 16 June to 7 August 2015, III). The general characteristics of NPF events were calculated on the basis of particle size distribution, and factors affecting the occurrence of NPF were mainly discussed by analyzing condensation sink (CS), sulfuric acid, trace gases concentration and meteorological parameters. In addition, one typical NPF event was specifically investigated to further explore the unusual NPF occurrence during haze episodes.

## 2 Methodology

### 2.1 Site description

The observations were conducted at the summit of Mt. Tai (36.25 °N, 117.1 °E, 1534 m ASL), located nearly the centre of Shandong Province, eastern China. Mt. Tai is one of the highest mountains near the East China Sea on the transport path of the Asian continental outflow (Li et al., 2011), adjacent to the Bohai Sea (B-S) and Yellow Sea (Y-S). The field site is just at the summit of Mt. Tai, and its surroundings are dominated by dense vegetation and mountains with few anthropogenic sources. The nearest small city, Tai'an (population: 670,000), is located about 15 km away to the south and southeast. The city of Ji'nan (population: 2,800,000), capital of Shandong Province, is 60 km to the north. During daytime, the summit of

Mt. Tai reaches close to the top of the planetary boundary layer (PBL), and the observation site is representative of the region (Zhang et al., 2014; Sun et al., 2016). All the instruments were installed inside a large container, sampling through short inlet tubes outside the container at a height of about 3 m above the ground level.

## 2.2 Measurement techniques

Two types of particle size distribution instruments, namely neutral cluster and air ion spectrometer (NAIS) and wide-range particle spectrometer (WPS), two gas monitors ($SO_2$ and $O_3$), an instrument for mass concentration of $PM_{2.5}$ and a monitor for chemical composition in $PM_{2.5}$ were used in this study. In addition, meteorological parameters including air temperature (T), relative humidity (RH), wind speed (WS), wind direction (WD) and visibility were also recorded in real time.

The NAIS is a multichannel nanometer aerosol instrument which can measure the size distribution of aerosol particles and ions (charged particles and cluster ions) of both polarities simultaneously. The aerosol particle distribution of NAIS is in the size range of 2-40 nm, and the ion distribution is in the electric mobility range of 0.0013-3.2 $cm^2$ $V^{-1}$ $s^{-1}$ (equivalent to particle Millikan diameters of 0.8-40 nm). The instrument consists of two multichannel electrical mobility analyzer columns, one for each polarity. The aerosols are classified according to electrical mobility and measured with an array of twenty-one electrometers per column. In this study, the total time of each measurement cycle was set at 5 min, comprising of sampling intervals as follows: particles 120 s, ions 120 s and offset 60 s.

The WPS is a high-resolution aerosol spectrometer which combines a differential mobility analyzer (DMA), a condensation particle counter (CPC) and a laser light scattering (LPS). The diameter range of WPS was from 10 to 10,000 nm, and 48 channels were used in the DMA and 24 channels were used for the LPS. The one scan time for the entire size range was set to 5 min.

The concentration of $SO_2$ in the atmosphere was measured using a pulsed ultraviolet fluorescence analyzer (Model 43C, Thermo Electron Corporation-TEC), and $O_3$ was measured using an ultraviolet photometric analyzer (Model 49C, TEC). Mass concentration of $PM_{2.5}$ was detected by a monitor utilizing a combination of beta attenuation and light scattering technology (Model 5030 SHARP Monitor, Thermo Fisher Scientific), and chemical composition in $PM_{2.5}$ was measured by a Monitor for Aerosols and Gases (MARGA, ADI20801, Applikon-ECN, Netherlands). Meteorological data were obtained in real time with an automatic meteorological station (MILOS520, Vaisala, Finland).

## 2.3 Data analysis

### 2.3.1 Formation rate, growth rate and condensation sink

In this study, particles in the size range of 3-20 nm was regarded as nucleation particles, and the formation rate of nucleation mode particles, $J_{3-20}$, can be expressed (Dal Maso et al., 2005) as:

$$J_{3-20} = \frac{dN_{3-20}}{dt} + F_{CoagS} + F_{growth} \tag{1}$$

where $dN_{3-20}/dt$ is the net rate of increased nucleation mode particles, $F_{CoagS}$ is the coagulation loss and $F_{growth}$ is the loss of particles growing out of size range. In our observation, the $F_{growth}$ term could be neglected because particles growing beyond 20 nm before formation ended was relatively rare. In addition, the formation rate of 3 nm particles, $J_3$, was also calculated from the NAIS data (Sihto et al., 2006; Kulmala et al., 2012) using the equation:

$$J_3 = \frac{dN_{3-6}}{dt} + \text{CoagS}_{Dp=4\ nm} \cdot N_{3-6} + \frac{1}{3\ nm}\text{GR}_{3-6} \cdot N_{3-6} \tag{2}$$

Here $\text{CoagS}_{DP=4\ nm}$ represents the coagulation sink of 4 nm particles, an approximation for the interval of 3-6 nm particles. $\text{GR}_{3-6}$ and $N_{3-6}$ denote the particle growth rate and particle number concentration between 3 and 6 nm, respectively.

The particle growth rate, GR, was determined by the maximum concentration method (Kulmala et al., 2012), that is:

$$\text{GR} = \frac{\Delta D_m}{\Delta t} = \frac{D_{m2} - D_{m1}}{t_2 - t_1} \tag{3}$$

where $D_{m1}$ and $D_{m2}$ are the geometric median diameters of representative particles at the start time $t_1$ and the end time $t_2$, respectively.

Condensation sink, CS, determines the rate of molecules condensing on the pre-existing aerosols, and it is given by (Dal Maso et al., 2005; Kulmala et al., 2001):

$$\text{CS} = 2\pi D \sum_i \beta_{Mi}\, Dp_i\, N_i \tag{4}$$

where $D$ is the diffusion coefficient for sulfuric acid, and $\beta_M$ is the size-dependent transitional correction factor.

### 2.3.2 Sulfuric acid proxy

Direct measurement of gas-phase sulfuric acid was not available in this study. Instead, the predictive proxy for sulfuric acid

($[H_2SO_4]$) could be estimated based on the solar radiation, $SO_2$ concentration, CS and RH (Mikkonen et al., 2011):

$$[H_2SO_4] = 8.21 \cdot 10^{-3} \cdot k \cdot \text{SR} \cdot [SO_2]^{0.62} \cdot (\text{CS} \cdot \text{RH})^{-0.13} \tag{5}$$

Here $k$ is a temperature-dependent reaction rate constant, and the solar radiation is estimated from Hybrid Single Particle Lagrangian Integrated Trajectory (HYSPLIT) Model in Air Resources Laboratory. The absolute values of solar radiation from HYSPLIT may involve some error due to uncertainty of solar radiation in the atmosphere, but its diurnal variation

pattern is acceptable.

### 3 Results and discussion

### 3.1 Classification and characteristics of NPF events

Basically, a NPF event could be defined as a distinct burst of new nucleation mode particles and subsequent growth of particles to larger size over a period of time (Dal Maso et al., 2005; Hallar et al., 2011; Wang et al., 2014a; Xiao et al., 2015).

For the Mt. Tai observations in this study, neutral particles generally accounted for more than 95 % of the total particles during NPF events, so a detailed discussion of ions will not be the focus in this paper.

The data presented in this study covered three campaigns from 25 July to 24 August 2014 (I), 21 September to 9 December 2014 (II), and 16 June to 7 August 2015 (III) at the summit of Mt. Tai. Observations over 164 days showed that the NPF events occurred on 66 days, corresponding to an occurrence frequency of 40 %. NPF events were observed right through the measurement campaigns with a highest occurrence frequency of 56 % during campaign II (during the other two campaigns the frequency was averagely 21 %). The difference may be attributed to the rainy/foggy conditions that prevailed during the campaigns I and III which have hindered NPF. In this study, we defined the observed start time of NPF events based on the significant enhancement of particle number concentration between 3 and 6 nm, $N_{3-6}$. The analysis showed that approximately 95 % of NPF events were initiated at 8:00-11:00 LT (local time) at the summit of Mt. Tai, which is in good agreement with many previous reports in China (Guo et al., 2012; An et al., 2015; Kulmala et al., 2016; Hao et al., 2015). The previous researches pointed out that this time period corresponded to the intensive photochemical activities, leading to potential precursors for NPF (Hallar et al., 2011; Guo et al., 2012). Table 1 lists the calculated parameters of all the NPF events observed at the summit of Mt. Tai, such as formation rate of nucleation mode particles, formation rate of 3 nm particles, growth rate, condensation sink, average sulfuric acid proxy concentration in the early morning (generally corresponding to time period of 6:00-9:00 LT on Mt. Tai), $SO_2$ concentration (6:00-13:00 LT), and $O_3$ concentration (6:00-13:00 LT). Table 2 summarizes the averages, medians, 25th percentiles, 75th percentiles, minima and maxima of these parameters on the basis of the Table 1. In Table 3, it compares the characteristics of NPF on Mt. Tai in the study and some other recent researches in China.

The net increase rates and formation rates of nucleation mode particles on Mt. Tai were in the range of 0.96-48.52 cm$^{-3}$ s$^{-1}$ and 1.10-57.43 cm$^{-3}$ s$^{-1}$, respectively. Coagulation loss averagely accounted for 24.6 % of the nucleation mode particle formation. The maximum values of the net increase rate, formation rate and $SO_2$ concentration all occurred on the same day-3 December 2014, and the $SO_2$ concentration on this day was 12.9±9.6 ppb. The formation rates $J_3$ varied from 0.82 to 25.04 cm$^{-3}$ s$^{-1}$, and the median, 25th percentile, and 75th percentile were 6.15, 3.31, and 9.41 cm$^{-3}$ s$^{-1}$, respectively. On 3 December 2014, $J_3$ also peaked, showing that the NPF event was controlled by sulfuric acid. As shown in Table 3, the particle formation rate at the summit of Mt. Tai was significantly larger than 0.97-10.2 cm$^{-3}$ s$^{-1}$ on the hillside of Mt. Tai Mo Shan and 0.09-0.30 cm$^{-3}$ s$^{-1}$ on the top of Mt. Huang (Guo et al., 2012; Zhang et al., 2016), but smaller than the results in Beijing and Shanghai (Xiao et al., 2015; Wang et al., 2015). In addition, the observed formation rate on Mt. Tai was a little larger than the rural/suburban environments included in Table 3 (Liu et al., 2008; Yue et al., 2013; Qi et al., 2015), which was likely associated to intensive precursor transport in the region (eastern China) and enhanced photochemical activity at the summit of Mt. Tai.

Growth rates GR$_{3-20}$ at the summit of Mt. Tai ranged from 0.58 to 7.76 nm h$^{-1}$, and the median, 25th percentile and 75th percentile were 1.55, 1.15 and 2.51 nm h$^{-1}$, respectively. Growth rate on Mt. Tai was comparable with some mountain observations such as 1.5-8.4 nm h$^{-1}$ at Mt. Tai Mo Shan, 1.42-4.53 nm h$^{-1}$ on Mt. Huang, and 0.8-3.2 nm h$^{-1}$ on Mt. Daban

(Du et al., 2015; Guo et al., 2012; Hao et al., 2015; Zhang et al., 2016). Growth rates observed at the rural/suburban/urban sites behaved lager than these mountain observations shown in Table 3 (Yue et al., 2013; Liu et al., 2008; Gao et al., 2011; Qi et al., 2015; Xiao et al., 2015), suggesting that relatively clean mountain environment might not contain insufficient vapors for newly formed particle growth.

## 3.2 Factors affecting the occurrence of NPF

Favorable conditions for NPF events vary according to different environments. In order to further explore the factors affecting the occurrence of NPF at the summit of Mt. Tai, multiple parameters were compared between NPF days and non-NPF days. Figure 1 shows the average diurnal variations of the condensation sink, sulfuric acid proxy, sulfur dioxide, ozone, temperature, and relative humidity during NPF days and non-NPF days over all the campaigns.

### 3.2.1 Effects of condensation sink and sulfuric acid on NPF

In principle, observation of an NPF event is dependent on the competition between the relevant sinks and sources. Newly formed particles are easily scavenged by larger pre-existing particles, leading to their continual reduction. On the other hand, a sufficiently high concentration of low volatility vapors can contribute to persistent nucleation. Therefore, CS and precursor vapors are the key controlling factors associated with NPF. Accordingly, as reported by Wang et al. (2011) and Guo et al. (2012), lower CS and higher precursor concentrations are favorable for the occurrence of NPF events. However, not all the environments meet the above two conditions simultaneously, and NPF events could still be observed in many other such environments. For example, Kulmala et al. (2016) reported that NPF events were observed in some polluted Chinese megacities with high aerosol loadings, and Zhu et al. (2014) showed that NPF occurred in Qingdao where a high concentration of gaseous pollutants might offset the effect of large CS. In a semi-rural location in India, Kanawade et al. (2014) also demonstrated that NPF were not limited by CS alone. Therefore, a detailed scientific analysis of the sinks and sources in specific atmospheric environments is of great important.

The hourly average CS on non-NPF days was always higher than that on NPF days (Fig. 1a), being $2.0\pm0.5\times10^{-2}$ s$^{-1}$ and $1.4\pm0.5\times10^{-2}$ s$^{-1}$, respectively. The result indicates that the occurrence of NPF at the summit of Mt. Tai is limited by the lower CS condition. The general diurnal tendencies of CS on NPF days and non-NPF days were consistent, showing the trough in the morning and the peak in the late afternoon or at night. The difference in the hourly values between NPF days and non-NPF days was the maximum in the morning and then decreased after 14:00 LT, leading to minimal values during 18:00-20:00 LT. The phenomena could be attributed to the new particle formation and subsequent growth patterns during NPF days. The CS on NPF days varied from $0.1\times10^{-2}$-$28.4\times10^{-2}$ s$^{-1}$, corresponding to the median, 25th percentile, and 75th percentile were $0.9\times10^{-2}$ s$^{-1}$, $0.5\times10^{-2}$ s$^{-1}$ and $1.7\times10^{-2}$ s$^{-1}$, respectively. This result was much larger than $0.5\times10^{-3}$-$3.5\times10^{-3}$ s$^{-1}$ in Hyytiälä, Finland (Dal Maso et al., 2005), but significantly lower than at many locations in China such as

$0.6\times10^{-2}$-$8.4\times10^{-2}$ s$^{-1}$ in Beijing, $0.9\times10^{-2}$-$3.9\times10^{-2}$ s$^{-1}$ in Nanjing, $0.9\times10^{-2}$-$5.3\times10^{-2}$ in Qingdao, and $1.0\times10^{-2}$-$6.2\times10^{-2}$ s$^{-1}$ in Hong Kong (Zhang et al., 2011; Gao et al., 2012; Guo et al., 2012; An et al., 2015; Herrmann et al., 2014; Zhu et al., 2014). Overall, the environment at the summit of Mt. Tai is relatively clean with low particle loadings.

Gas-phase sulfuric acid has been identified as an important precursor for the nucleation process. Direct emission for sulfuric acid at the summit of Mt. Tai is negligible, thus the photochemical reactions of SO$_2$ would be the significant source for sulfuric acid in the atmosphere. In this study, the average sulfuric acid proxy concentrations between 6:00-9:00 LT of all the NPF days were in the range of $0.52\times10^6$-$25.7\times10^6$ cm$^{-3}$. The sulfuric acid proxy concentration on both NPF days and non-NPF days peaked near the mid-day (Fig. 1b), which was consistent with the peak time of solar radiation. The maximum hourly average sulfuric acid proxy concentration on NPF days ($5.9\times10^7$ cm$^{-3}$) was about 60 % higher than that on non-NPF days ($3.7\times10^7$ cm$^{-3}$). In the early morning, the average sulfuric acid proxy concentration on NPF days was $5.23\times10^6$ cm$^{-3}$, comparable with $4.1\times10^6$ cm$^{-3}$ in Beijing but much lower than $2.3\times10^7$-$6.4\times10^7$ cm$^{-3}$ in Shanghai and $6.6\times10^7$-$7.8\times10^7$ cm$^{-3}$ in Nanjing (Wang et al., 2014b; Wang et al., 2015; Xiao et al., 2015). More polluted observation environments generally have the elevated CS, and therefore will require a higher concentration of condensable vapors to initiate the nucleation. This is indirectly reflected in many oversea studies in which NPF events are observed at clean or moderately-polluted sites in the presence of lower sulfuric acid concentrations (Dal Maso et al., 2005; Boy et al., 2005). Apart from other precursor species, lower initial sulfuric acid proxy concentration can be partly explained by the relatively lower CS at the summit of Mt. Tai.

The relationships between sulfuric acid, CS and the $N_{3-6}$ in the early morning are shown in Fig. 2. Most of data points denoted the CS on non-NPF days were situated in the intermediate region ($\sim1.0\times10^{-2}$ s$^{-1}$), whereas approximately 80 % of CS on NPF days was less than $1.0\times10^{-2}$ s$^{-1}$. It is expected that the left side of the chart (small CS) is more favorable for the particle nucleation than the right side (large CS). The distribution of sulfuric acid proxy concentration data points on the vertical axis showed no difference between NPF days and non-NPF days, which is consistent with the trend in Fig. 1b during 6:00-9:00 LT. This phenomenon might be explained by the intensive photochemical activities and high oxidation capacity due to the enhanced solar radiation in the morning at the summit of Mt. Tai, which gave rise to a sufficiently high vapor concentration for the nucleation burst. Therefore, sulfuric acid concentration sometimes is not the major limiting factor for the onset of NPF at the summit of Mt. Tai.

It was noteworthy that $N_{3-6}$ showed a significant correlation with the sulfuric acid proxy concentration on NPF days (Fig. 2a). In general, $N_{3-6}$ was less than 500 cm$^{-3}$ when the sulfuric acid proxy concentration was lower than $1.0\times10^6$ cm$^{-3}$, and it often exceeded 800 cm$^{-3}$ when the sulfuric acid proxy concentration was greater than $1.0\times10^6$ cm$^{-3}$. Further investigation found that sulfuric acid proxy concentration showed a clear positive correlation with $N_{3-6}$ on many NPF days. In contrast, Fig. 2b did not exhibit the similar relationship between $N_{3-6}$ and the sulfuric acid proxy concentration on non-NPF days, and the colors of data points ($N_{3-6}$) were almost evenly distributed across the sulfuric acid proxy concentration as shown on the vertical axis.

Being the most critical nucleation precursor, sulfuric acid was associated with the freshly nucleated particles. As above, the positive correlation between sulfuric acid proxy concentration and $N_{3-6}$ at the summit of Mt. Tai was in accordance with earlier reports (Kulmala et al., 2006; Wang et al., 2011; Guo et al., 2012). As an example in Fig. 3, $N_{3-6}$ reflected a best relationship ($R^2$ =0.975) with sulfuric acid proxy concentration during the NPF event between 6:00 and 14:00 LT on 14 October 2014. After 14:00 LT, $SO_2$ concentration increased sharply (change from 2.6 ppb at 14:00 LT to 19.1 ppb at 15:00), resulting in a lack of correlation between them. In principle, the increase of sulfuric acid concentration should take place earlier than the increase in $N_{3-6}$. However, there were some NPF days with zero or negative time delay in this study, such as on 14 October 2014 in Fig. 3. Wang et al. (2011) suggested that the pre-formed nucleation mode particles and rapid particle growth might account for such zero or negative time delay.

**3.2.2 Effects of sulfuric dioxide and ozone on NPF**

As shown in Fig. 1c, almost all the hourly average $SO_2$ concentrations on NPF days were higher than the corresponding values on non-NPF days (except for the slightly lower values at 1:00 LT and 3:00 LT), which indicated that NPF was really favorable to the high $SO_2$ concentration. The average $SO_2$ concentrations between 6:00-13:00 LT on NPF days and non-NPF days were 3.2 ppb and 2.6 ppb, respectively. Photochemical reactions of $SO_2$ are the major source for sulfuric acid at the summit of Mt. Tai, thus the higher $SO_2$ concentration can increase the possibility of rich precursors for NPF. For example, it was noteworthy that the temperature suddenly dropped from 1.3 ℃ to -9.4 ℃ on 30 November 2014 in Fig. 4. After several days, an exceptionally high $SO_2$ concentration was observed (7.1 ±7.2 ppb) at the summit of Mt. Tai (marked in violet block), and frequent NPF events occurred during this period. A possible reason for this observation might be the entrainment of anthropogenic pollutants (such as $SO_2$) from coal or biomass burning in the upwind region (Li et al., 2015a; Li et al., 2015b).

Some of NPF events did not show sensitivity to $SO_2$ concentration during our observations. For example, on 21 September 2014 a non-NPF event occurred under simultaneously high $SO_2$ concentration (9.7 ±7.0 ppb during 6:00-13:00 LT) and CS ($4.0 \pm 1.2 \times 10^{-2}$ s$^{-1}$ during 6:00-13:00 LT) conditions. The average $PM_{2.5}$ concentration on this study was 61 μg m$^{-3}$. It appears that the sink dominated the inter-competition between sink and source during the NPF event, which possibly limited the particle nucleation. Previous studies have also reported the above similar phenomena. Song et al. (2010) attributed the weak dependence between $SO_2$ concentration and NPF events to the intimate coupling between source gases and pre-existing aerosols in South Korea. Therefore, other species accompanying $SO_2$ emission need to be taken into account. Of course, the coupling of $SO_2$ and the accompanying species at the summit of Mt. Tai may be weakened significantly due to variable air mass transport.

The oxidation capacity and photochemical activities in the atmosphere are affected by the atmospheric $O_3$, directly reacting with related species such as VOCs, and indirectly affecting sulfuric acid formation via hydroxyl and hydroperoxy radicals (Berndt et al., 2010; Gómez Martń et al., 2013; Sorribas et al., 2015; Guo et al., 2012). The average $O_3$ concentrations on NPF

days and non-NPF days were 40 ppb and 47 ppb, respectively, and all the hourly average $O_3$ concentrations on NPF days were lower than that on non-NPF days (Fig. 1d). Previous reports showed that elevated $O_3$ concentration was beneficial to the occurrence of NPF (An et al., 2015; Guo et al., 2012; Zhang et al., 2016; Huang et al., 2016), but our results did not directly show this effect. It is known that $O_3$ is usually contributed by anthropogenic emissions on the ground, such as $NO_2$. Elevated $O_3$ concentration enhances the atmospheric oxidation capacity, but it can increase some negative accompanying pollutants simultaneously, such as aged particles. During NPF events at the summit of Mt. Tai, the positive influence of the increase in $O_3$ may not always offset the negative influence of accompanying pollutants. Therefore, it may be concluded that a relatively low $O_3$ environment could favour of NPF at the summit of Mt. Tai in this study. The diurnal variation of $O_3$ concentration at the summit of Mt. Tai had two prominent features - a small trough in the early morning resulting from dry deposition and a broad peak in the afternoon due to formation by solar radiation. This may explain why the hourly difference values of $O_3$ between NPF days and non-NPF days in Fig. 1d decreased after 10:00 LT.

### 3.2.3 Effects of meteorological conditions on NPF

Favorable meteorological conditions can promote the occurrence of NPF when precursors are insufficient in the atmosphere (Song et al., 2010). In this study, approximately 90 % of NPF events occurred during clear or partial cloudy daytime, suggesting the strong influence of solar radiation on NPF. During the three campaigns, the air temperature of all NPF days varied from -11.8 to 22.1 ℃, and the daily temperature profiles were characterized by the expected cosine form of curve, and daily average temperature generally oscillated less than 10 ℃ within a day. The temperature at the summit of Mt. Tai exhibited a clear seasonal behavior. As observed in Fig. 1e, the hourly average temperatures on non-NPF days (10.7±1.1 ℃) were always higher than that on NPF days (7.0±1.8 ℃), indicating that NPF at the summit of Mt. Tai preferred relatively low temperature. This is in good agreement with previous observations at Mt. Tai Mo Shan (Guo et al., 2012), Mt. Huang (Zhang et al., 2016), PUY in France (Rose et al., 2015) et al., which all reported the relationship between lower temperature and NPF events. Guo et al. (2012) explained that the favorable low temperature could enhance the nucleation of sulfuric acid and water vapor. In addition, in Young et al. (2007) it showed that the effect of lower temperature during NPF events was be contributed by atmospheric vertical convection. Overall, NPF events at the summit of Mt. Tai seemed to occur more readily at lower temperature.

The RH of all the NPF days ranged from 22 % to 95 %, and diurnal variation of RH was inversely correlated with the solar radiation. The hourly average RH on NPF days (63±5 %) was always much lower than the corresponding value on non-NPF days (88±2 %), and the maximum difference between the two curves in Fig. 1f was about 30 %. An anti-correlation between the NPF and RH can be identified at the summit of Mt. Tai in the study, which is in agreement with the results in Beijing, Nanjing, Hong Kong, and Mt. Huang (An et al., 2015; Wang et al., 2014a; Shen et al., 2016; Zhang et al., 2016; Guo et al., 2012). The actual role of RH on NPF is still controversial and has not been resolved. Hamed et al. (2011)

indicated that the RH affected the source of NPF via decreasing solar radiation under the high RH conditions. In contrast, simultaneous increasing CS (sink) under the high RH conditions has been suggested as an explanation for the negative effect of the RH (Hamed et al., 2011; Guo et al., 2012).

A wide range of wind speed was on NPF days at the summit of Mt. Tai, varying from 0.2 m s$^{-1}$ to 7.8 m s$^{-1}$. Throughout the three campaigns, the dominant wind directions at the summit of Mt. Tai were east and west, with the main directions being 40°-110° and 220°-300°. North and south wind directions were generally rare in this study. Compared with non-NPF days, the wind direction on NPF days had a more narrow range in the east-southeast (85°-110°) and west-southwest directions (250°-300°), as shown in Fig. 5. The CS at the summit of Mt. Tai changed with the wind directions, and the average CS in the wind directions between 40° and 110° ($2.0\times10^{-2}$ s$^{-1}$) were almost twice as high as the wind directions between 220° and 300° ($1.1\times10^{-2}$ s$^{-1}$). It is to be noted that the CS showed a relatively low value when the wind came from the east-southeast direction, partly explaining why the occurrence of NPF events were more frequent when the wind was from this particular direction. The SO$_2$ concentrations were almost evenly distributed with wind directions, except for the west-southwest direction which corresponded to an elevated SO$_2$ concentration of 4.3 ppb. For the adjacent wind directions between 220° and 350°, the daily average SO$_2$ concentrations were just 2.0 ppb. This suggests that the gas sources in the west-southwest direction may be contributing to an increase in the probability of NPF occurrence to a certain extent.

### 3.2.4 Effects of air mass transport on NPF

To characterize the influence of the long range air mass transport on NPF at the summit of Mt. Tai, air mass backward trajectories for 72 h at 6:00 LT at 1535 m ASL were simulated using the HYSPLIT model developed by the National Oceanic and Atmospheric Administration (NOAA) Air Resources Laboratory. Figure 6a illustrates the air mass backward trajectories of all the NPF days, and Fig. 6b shows for all the non-NPF days.

Based on the transport range and distance, the air mass backward trajectories were classified into three categories: continental air mass (red in Fig. 6), local air mass (green in Fig. 6) and maritime air mass (magenta in Fig. 6). The majority of transport pathways on NPF days were continental air mass, which accounted for 80 % of the total air masses. Continental air mass on NPF days mainly came from the northwest of observation site, and largely originated from Siberia passing over the long distance across Mongolia, Inner Mongolia, Shanxi Province, Hebei Province, Beijing et al.. The local and maritime backward trajectories on NPF days accounted for 8 % and 12 % of the total trajectories, respectively. The local air mass was mainly from surroundings such as Jinan, Nanjing, Zhengzhou et al. with the short routes, whereas the maritime air mass originated over the Bohai Sea (E-S), Yellow Sea (Y-S) and East China Sea (E-S). The percentages of the continental air mass, local air mass and maritime air mass on non-NPF days were 63 %, 12 % and 25 %, respectively. Overall, the local air mass accounted for the minimum proportion on both NPF and non-NPF days, and long range air mass transport had greatly important influence on NPF events at the summit of Mt. Tai. In addition, comparison between the Fig. 6a and Fig. 6b

showed that there was significant decreasing percentage of maritime air mass and shorter routes over ocean areas during NPF days, suggesting that the maritime air mass may not provide the conditions favorable for occurrence of NPF at the summit of Mt. Tai.

As above, the continental air mass dominated throughout our observations. According to its transported regions, polluted continental air mass (Type I) and clean continental air mass (Type II) were denoted in the study. Type I passed though the heavily polluted areas of Beijing, Hebei Province, Shanxi Province, Henan Province, and Shaanxi Province before air masses reaching the observation site, which could carry large amount of pollutants. Type II was either from the south China or transported over the Bohai Sea (B-S) and Yellow Sea (Y-S), so it represented the relatively clean air masses arriving in Mt. Tai. In this study, four-fifths of the continental air masses on NPF days were Type I, whereas Type I only accounted for two-fifths of the total continental air masses on non-NPF days. It seems plausible that air masses passing through the heavily polluted areas could increase the occurrence of NPF, which might be explained that transported air masses of Type I brought high level of anthropogenic species for NPF.

In order to further verify the speculation that Type I is in favor of NPF events, Fig. 7 illustrates the average chemical composition of $PM_{2.5}$ and $SO_2$ concentration in Type I and Type II during NPF days. A prominent increase of average $SO_2$ concentration was found in Type I, comparing between 3.9 ppb and 1.2 ppb. Higher $SO_2$ concentration suggested that the enhanced precursors could be carried to Mt. Tai when the air masses passed over the heavily polluted areas. In addition, the average mass concentrations of $PM_{2.5}$ in Type I and Type II were 33 μg m$^{-3}$ and 23 μg m$^{-3}$, respectively. The significantly elevated percentages of sulfate, ammonium and nitrate were showed in Type I, indicating that transported air masses really passed through polluted S-rich and N-rich areas which might provide potential species for NPF.

**3.3 Typical NPF events during haze episodes**

In Sect. 3.2, it mainly investigated the general conditions favorable for the occurrence of NPF at the summit of Mt. Tai. In fact, we have also observed NPF events occurring under some specific conditions, such as during haze episodes. According to the Chinese Meteorological Administration, the haze episode are defined when the atmospheric visibility is less than 10 km and the RH is less than 80 % simultaneously. In the atmosphere, haze episodes are always accompanied by elevated $PM_{2.5}$ concentrations. Higher $PM_{2.5}$ concentration implies the increased particle loadings, which suppresses the particle nucleation to a great extent. In this study, the hourly average $PM_{2.5}$ concentration on non-NPF days was always higher than the corresponding values on NPF days before 9:00 LT, comparing between 28 μg m$^{-3}$ and 22 μg m$^{-3}$ during 0:00-9:00 LT. An intensive investigation of NPF events occurring during haze episodes (denoted as NPF haze event below) is in this section.

In this study, four NPF haze events (26 July 2014, 17 October 2014, 18 October 2014, and 11 November 2014) were observed during the three campaigns, and the related parameters on these four days (denoted as NPF haze days below) were shown in Table 4. The $PM_{2.5}$ concentrations on NPF haze days were significantly higher than the average of all NPF days

observed at the summit of Mt. Tai, but potentially elevated precursor concentrations might exist simultaneously because of the high level of $SO_2$, $O_3$ and CO concentrations. The temperatures and RH were in accordance with the corresponding climates, and the wind speed on each NPF haze day varied considerably. It was noted that the start time of NPF haze events was around noon, an obvious time delay compared with the general start time of 8:00-11:00 LT of normal NPF events as shown in Set. 3.1. This phenomenon might suggest that the enhanced solar radiation at noon could be an important factor for the occurrence of NPF haze events. In order to further explore the characteristics of NPF haze events, one NPF event on 11 November 2014 was selected as a case study. The time series of the particle size distribution, meteorological parameters, trace gases, chemical composition and mass concentration of $PM_{2.5}$ on this day are illustrated in Fig. 8.

On 11 November 2014, an NPF event was clearly observed at about 11:30 LT. The visibility increased and the RH fell sharply as the fog episode dispersed. The haze episode was present before the nucleation, and thus the day was considered as a NPF haze day. The highest temperature was about 7 ℃ and the lowest RH was 66 % at noon. At about 10:00 LT, the clear increase in $SO_2$ concentration, decrease in $PM_{2.5}$ concentration and change in chemical composition of $PM_{2.5}$ were observed simultaneously. The observation site is located on the mountain top without any stationary source nearby, and the above changes therefore may suggest that there was another polluted air mass transported to the observation site, contributing to the NPF event to a certain extent. The average mass concentrations of sulfate, ammonium, nitrate, OC and EC were 4.1, 4.6, 4.1, 7.6 and 0.7 μg m$^{-3}$ during 6:00-9:00 LT, respectively. In contrast, their average concentrations were 13.1, 11.2, 11.1, 10.0 and 0.2 μg m$^{-3}$ in order during 11:00-18:00 LT. The ratio of sulfate increased significantly before and after 10:00 LT, suggesting that the new transported air mass was S ($SO_2$)-rich.

Air mass backward trajectories for 72 h at 1535 m ASL at 14:00, 12:00, 10:00, 8:00 and 6:00 LT on 11 November 2014 are shown in Fig. 9. It illustrated that the origin of transported air mass moved from eastern China (Jiangsu and Anhui Provinces, line A and B) to western China (line C, D and E) at about 10:00 LT, which was in agreement with the above analysis in Fig. 8. The latter air mass backward trajectories (line C, D and E) passed through the heavily polluted areas, such as Shanxi and Shaanxi Provinces, before reaching the observation site, leading to the potential increase of precursor concentrations.

**4. Conclusions**

Field observations of NPF at the mountain-top sites are scarce in China, and the results of such studies could significantly contribute to atmospheric aerosol pollution control. A comprehensive investigation of NPF was conducted at the summit of Mt. Tai (1534 m ASL), eastern China, from 25 July to 24 August 2014 (I), 21 September to 9 December 2014 (II) and 16 June to 7 August 2015 (III), using two types of size distribution instruments (NAIS and WPS) and mainly including two trace gases, multiple meteorological parameters, mass concentration and chemical composition of $PM_{2.5}$. During the 164 days, 66

NPF events were identified based on the particle size distribution, giving an occurrence frequency of 40 % overall. Most of NPF events were initiated at 8:00-11:00 LT, coinciding with the time period of the intensive photochemical activity. The $J_3$, $J_{3-20}$, and growth rates were in the range of 0.82-25.04 $cm^{-3} s^{-1}$, 1.10-57.43 $cm^{-3} s^{-1}$, and 0.58-7.76 nm $h^{-1}$, respectively. In comparison with other studies in China, particle formation rate at the summit of Mt. Tai was slightly larger than some rural/suburban environments, which probably was associated with the intensive air mass transport in eastern China and enhanced solar radiation at the summit of Mt. Tai. Instead, growth rate in this study showed lower value compared with some rural/suburban/urban environments of China, suggesting insufficient vapors contributing to particle growth rate on Mt. Tai.

NPF events generally occurred at lower CS, $O_3$ concentration, air temperature and RH conditions, whereas the respective $SO_2$ concentration was observed to be higher. The role of sulfuric acid concentration on the onset of NPF events is uncertain because there is no prominent superiority in sulfuric acid proxy concentration between NPF days and non-NPF days in the early morning. NPF events observed on Mt. Tai were more common when wind came from the east-southeast and west-southwest directions, which were mainly attributed to the relatively low CS in the east-southeast and elevated $SO_2$ concentration in the west-southwest. Backward trajectories were classified into continental, local and maritime air masses, and the majority was continental air mass. Maritime air mass may not provide the favorable conditions for NPF because of the significant decreasing percentage of maritime air mass on NPF days. The air masses passing through the heavily polluted areas (denoted as Type I) would have increased the frequency of NPF events, which was possibly explained by the high level of anthropogenic species carried from the polluted regions.

Four NPF events were observed during haze episodes, and elevated $PM_{2.5}$ and trace gases concentrations were observed simultaneously on these NPF haze days. A case study on 11 November 2014 showed that, the NPF event was probably caused by the enhanced solar radiation at noon and changed transported air mass which brought the potentially high precursor concentrations to the measurement site.

*Acknowledgements*. This work was supported by National Natural Science Foundation of China (No. 41375126), Mount Tai Scholar Grand (ts20120552), Cyrus Tang Foundation (No.CTF-FD2014001), Ministry of Science and Technology of China (SQ2016ZY01002231, 2014BAC22B01), and Marie Skłodowska-Curie Actions (H2020-MSCA-RISE-2015-690958).

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

**Figure and Table captions**

**Fig. 1.** Average diurnal variations of the condensation sink, sulfuric acid proxy concentration, trace gases ($SO_2$ and $O_3$), and meteorological conditions (T and RH) during NPF days and non-NPF days over all the campaigns at the summit of Mt. Tai.

**Fig. 2.** The relationships between sulfuric acid proxy, condensation sink and particle number concentration of 3-6 nm ($N_{3-6}$) in the early morning (6:00-9:00 LT) during NPF days (a) and non-NPF days (b). The data are hourly averages, and the marker colors represent the values of $N_{3-6}$ ($cm^{-3}$).

**Fig. 3.** The particle number concentration of 3-6 nm ($N_{3-6}$, blue) and sulfuric acid proxy concentration ([$H_2SO_4$], red) on 14 October 2014, fitting a good relationship ($R^2$ =0.975) between 6:00-14:00 LT during the NPF event.

**Fig. 4.** Time series during 10 November-9 December 2014, and the shaded areas represent the NPF days: (a) contour plot of particle number size distribution using NAIS data; (b) sulfur dioxide (blue) and ozone (red); (c) meteorological parameters, including wind speed (cyan), wind direction (green), temperature (magenta) and relative humidity (earth yellow); (d) visibility (yellow) and $PM_{2.5}$ concentration (purple), and the gray line is the boundary for 10 km and 75 μg $m^{-3}$.

**Fig. 5.** Wind rose plots of all the NPF days (a) and non-NPF days (b), and the wind speed and wind direction between 06:00 and 11:00 LT are included. Length of each spoke on the circle represents the probability of wind coming from a particular direction at the certain range of wind speed.

**Fig. 6.** Air mass back trajectories for 72 h at 6:00 LT at 1535 m ASL on all the NPF days (a) and non-NPF days (b), and the continental air mass, local air mass and maritime air mass are represented in red, green and magenta, respectively.

**Fig. 7.** The average chemical composition of $PM_{2.5}$ and $SO_2$ concentration in polluted continental air mass (Type I) and clean continental air mass (Type II) during NPF days

**Fig. 8.** Time series of particle size distribution, meteorological parameters, trace gases, chemical composition and mass concentration of $PM_{2.5}$ on 11 November 2014.

**Fig. 9.** Air mass backward trajectories for 72 h at 1535 m ASL at 6:00 (A), 8:00 (B), 10:00 (C), 12:00 (D) and 14:00 LT (E) on 11 November 2014.

**Table 1.** The calculated parameters of all the NPF events during the three campaigns, the minima and maxim are marked in blue and red, respectively

**Table 2.** Summary of averages, medians, 25th percentiles, 75th percentiles, minima and maxima for the calculated parameters on the basis of Table 1

**Table 3.** Comparison of NPF characteristics between Mt. Tai and other studies in China

**Table 4.** The parameters on four NPF haze days and the averages of all the NPF days during 6:00-18:00 LT

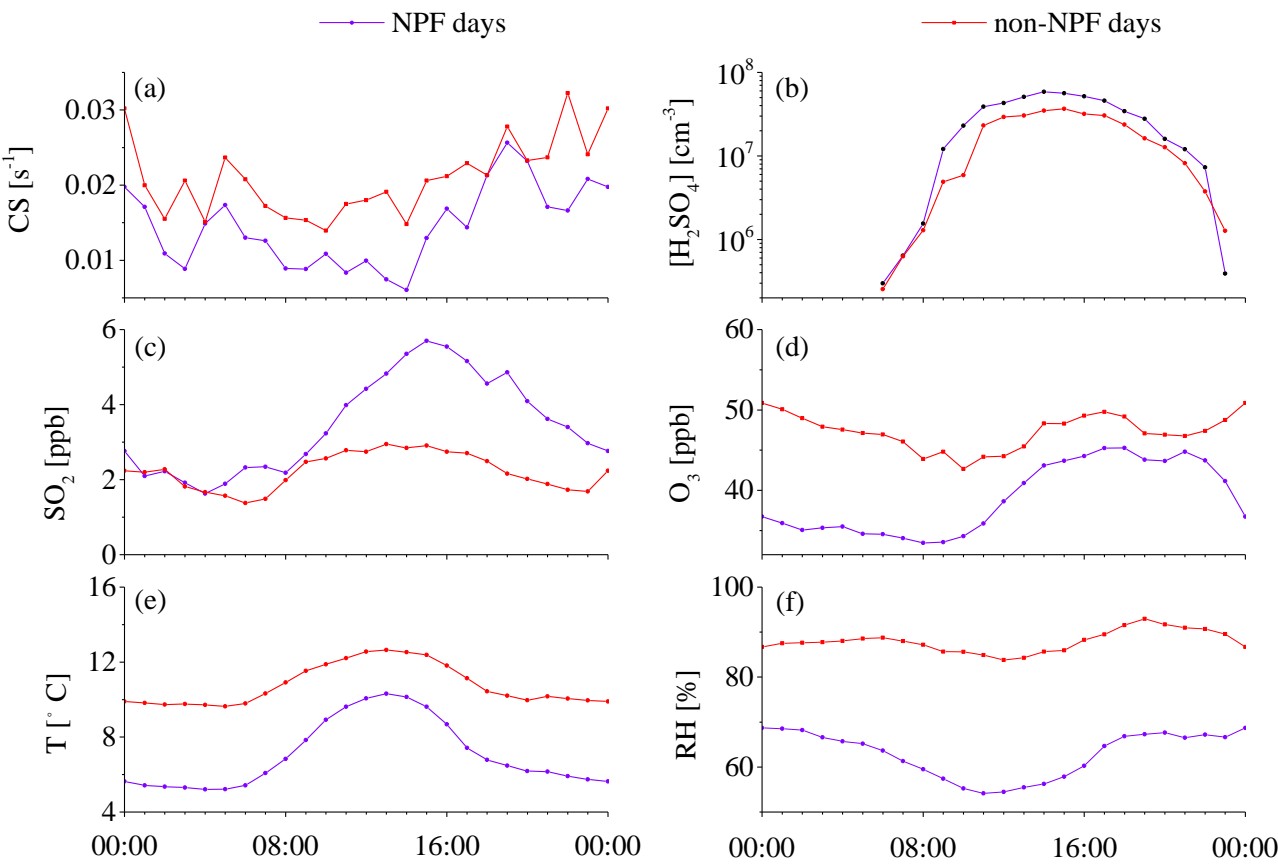

**Fig. 1.** Average diurnal variations of the condensation sink, sulfuric acid proxy concentration, trace gases (SO$_2$ and O$_3$), and meteorological conditions (T and RH) during NPF days and non-NPF days over all the campaigns at the summit of Mt. Tai.

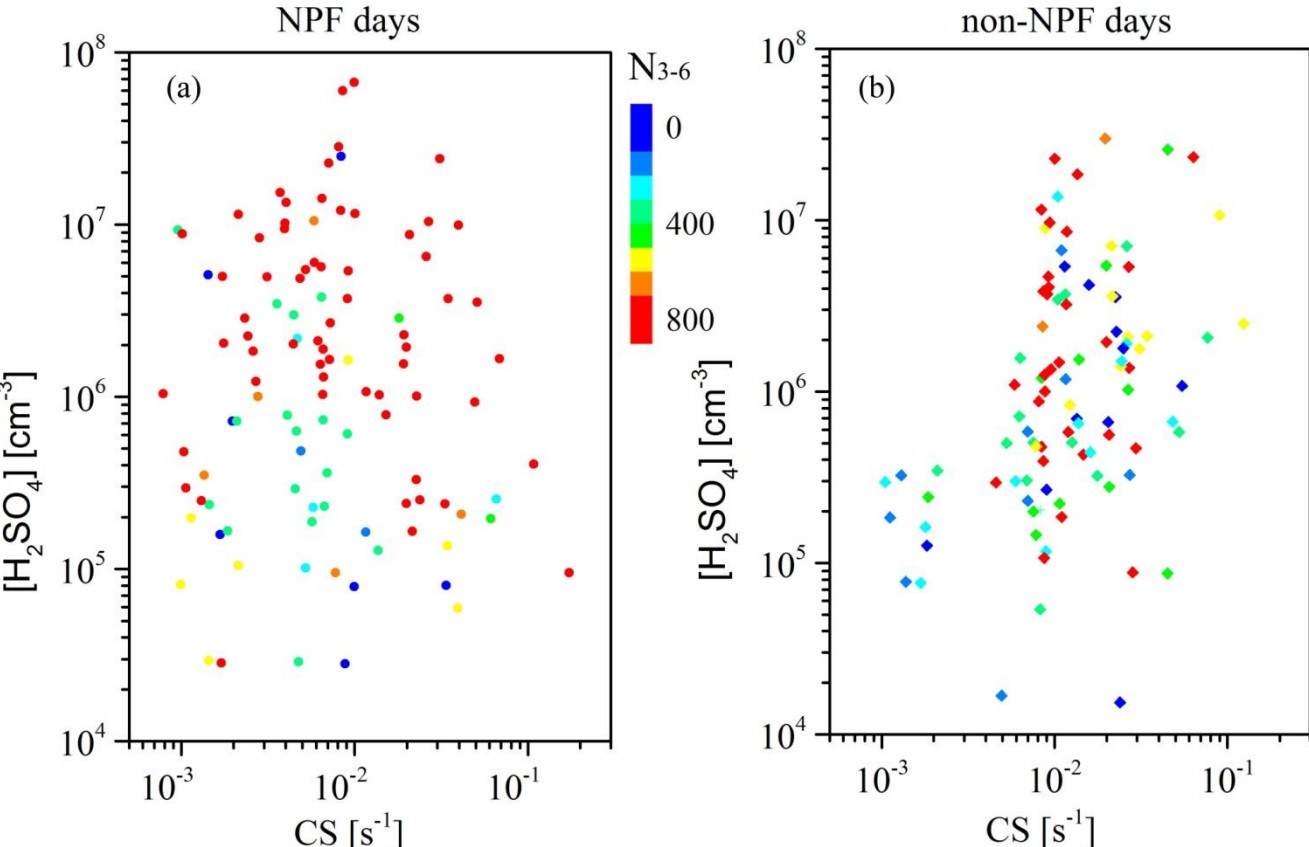

**Fig. 2.** The relationships between sulfuric acid proxy, condensation sink and particle number concentration of 3-6 nm ($N_{3-6}$) in the early morning (6:00-9:00 LT) during NPF days (a) and non-NPF days (b). The data are hourly averages, and the marker colors represent the values of $N_{3-6}$ (cm$^{-3}$).

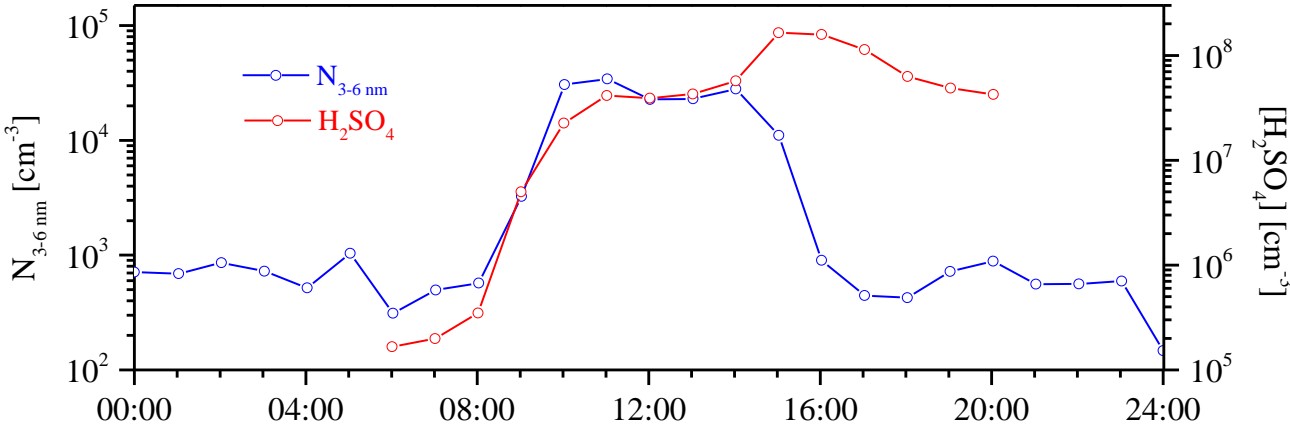

**Fig. 3.** The particle number concentration of 3-6 nm ($N_{3-6}$, blue) and sulfuric acid proxy concentration ([$H_2SO_4$], red) on 14 October 2014, fitting a good relationship ($R^2$ =0.975) between 6:00-14:00 LT during the NPF event.

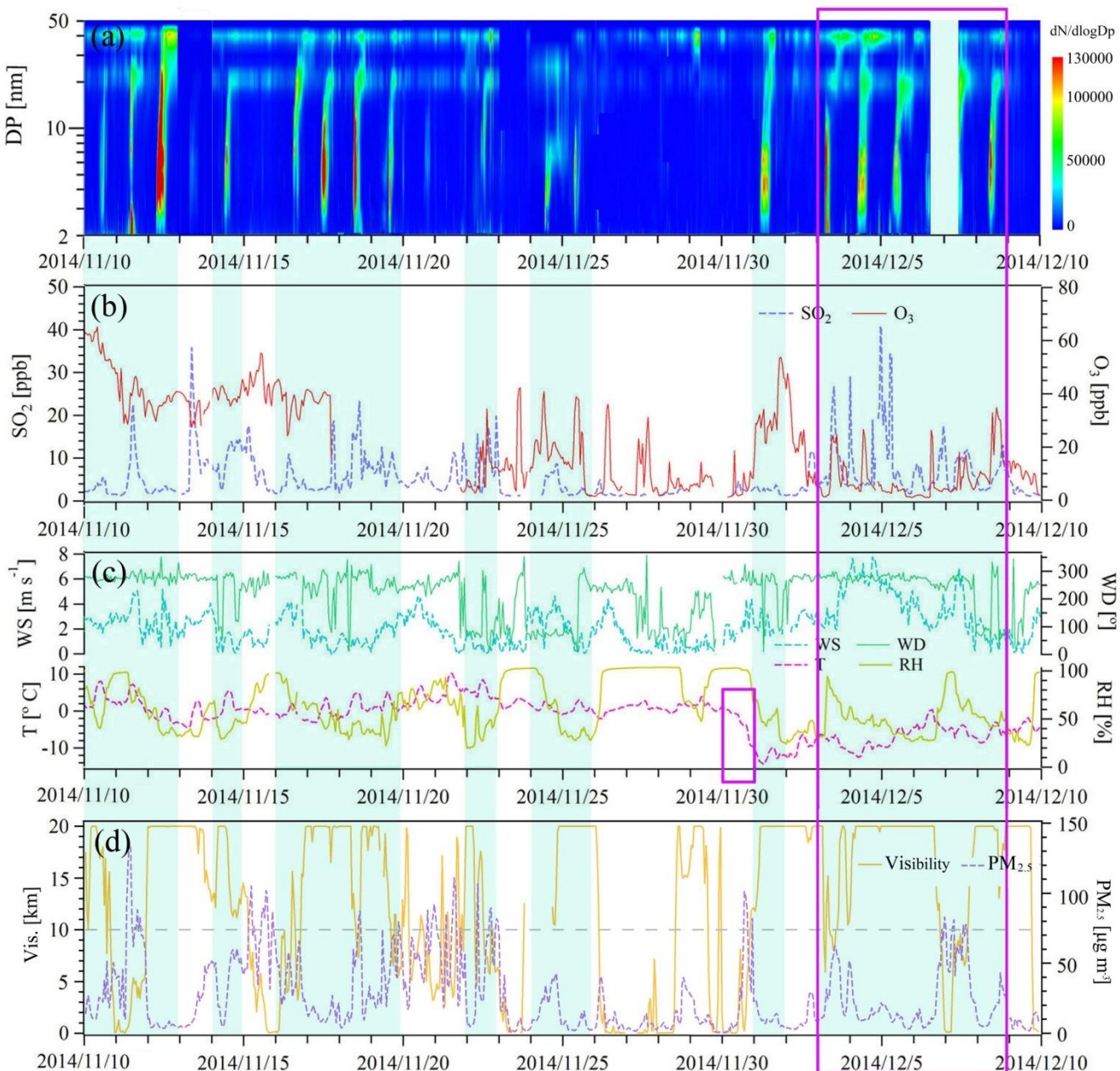

**Fig. 4.** Time series during 10 November-9 December 2014, and the shaded areas represent the NPF days: (a) contour plot of particle number size distribution using NAIS data; (b) sulfur dioxide (blue) and ozone (red); (c) meteorological parameters, including wind speed (cyan), wind direction (green), temperature (magenta) and relative humidity (earth yellow); (d) visibility (yellow) and $PM_{2.5}$ concentration (purple), and the gray line is the boundary for 10 km and 75 μg m$^{-3}$.

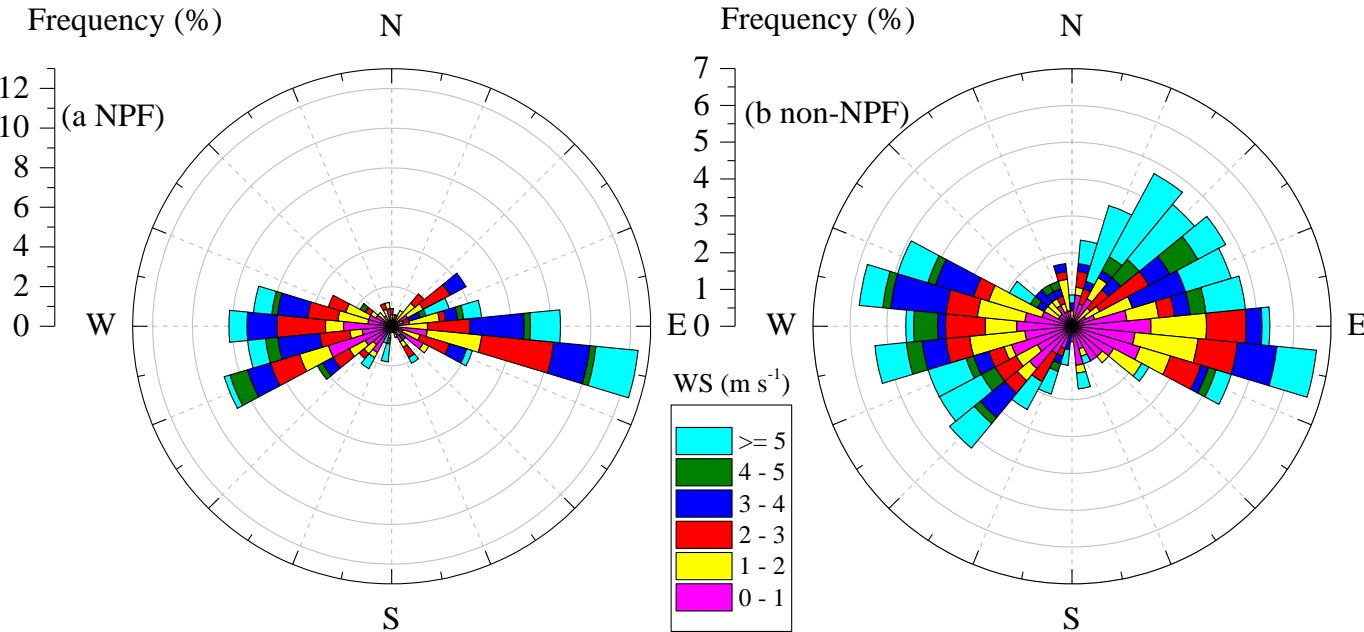

**Fig. 5.** Wind rose plots of all the NPF days (a) and non-NPF days (b), and the wind speed and wind direction between 06:00 and 11:00 LT are included. Length of each spoke on the circle represents the probability of wind coming from a particular direction at the certain range of wind speed.

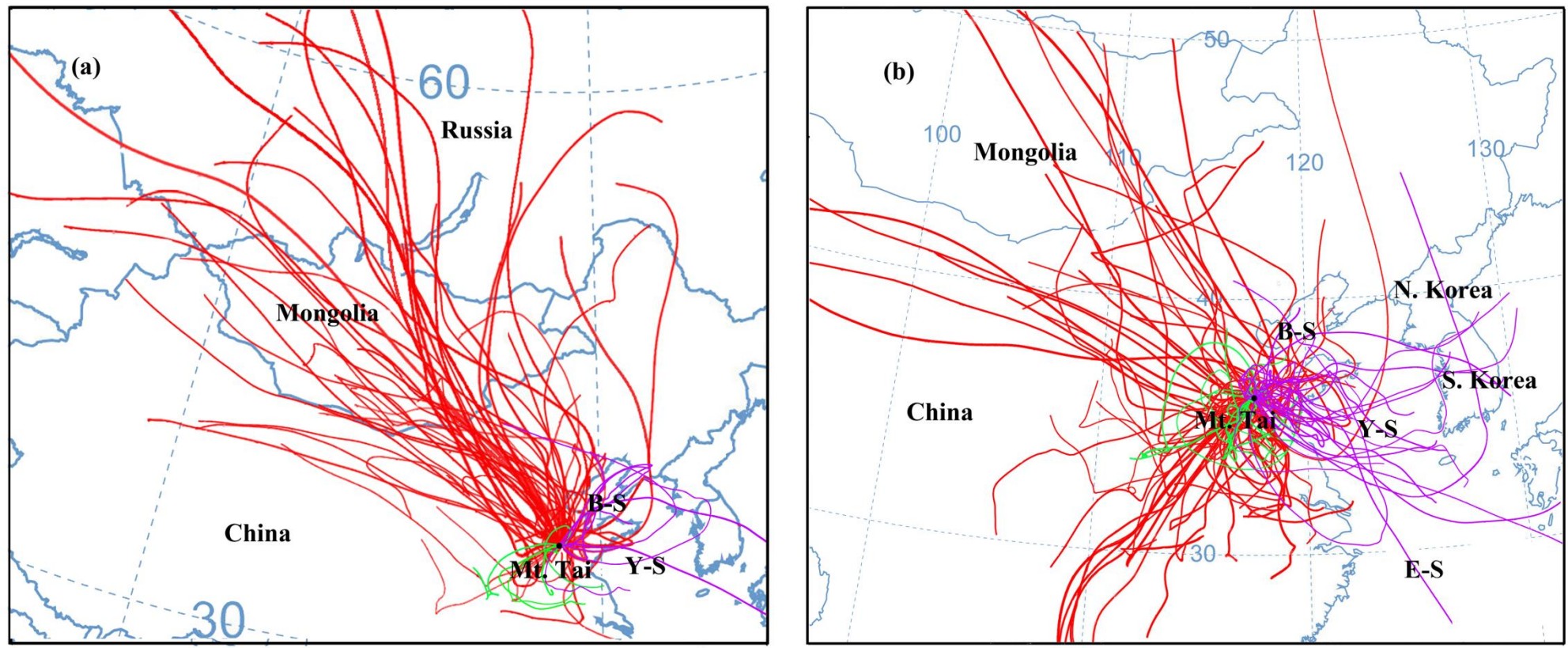

**Fig. 6.** Air mass back trajectories for 72 h at 6:00 LT at 1535 m ASL on all the NPF days (a) and non-NPF days (b), and the continental air mass, local air mass and maritime air mass are

represented in red, green and magenta, respectively.

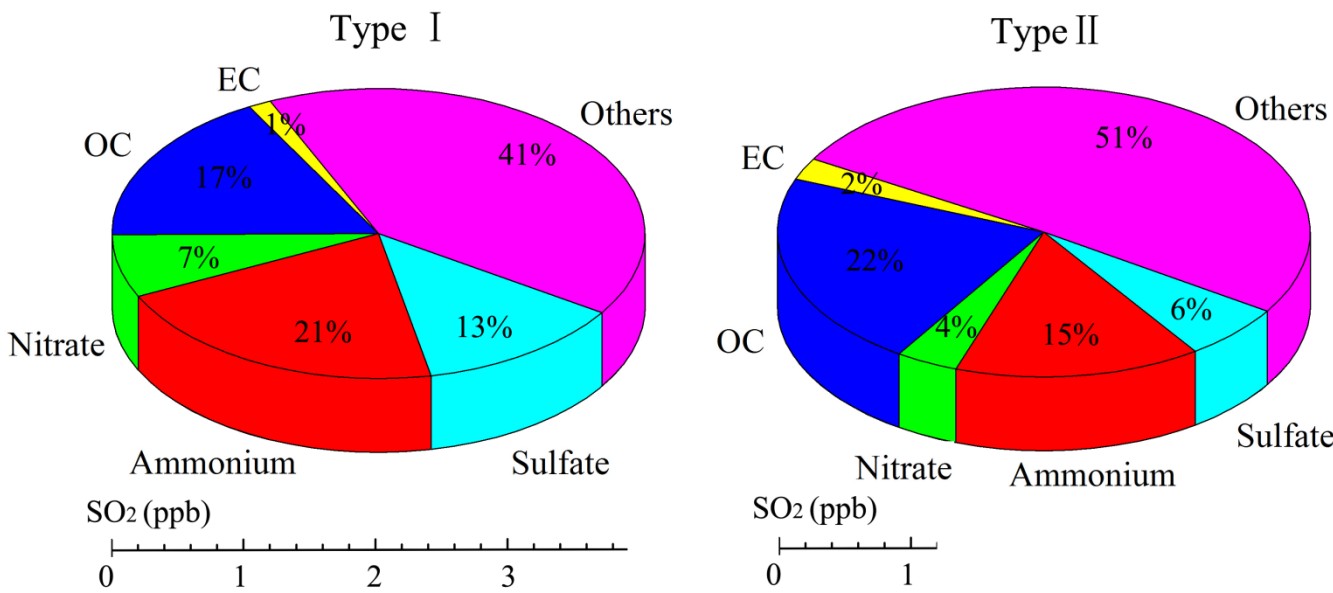

**Fig. 7.** The average chemical composition of PM$_{2.5}$ and SO$_2$ concentration in polluted continental air mass (Type I) and clean continental air mass (Type II) during NPF days

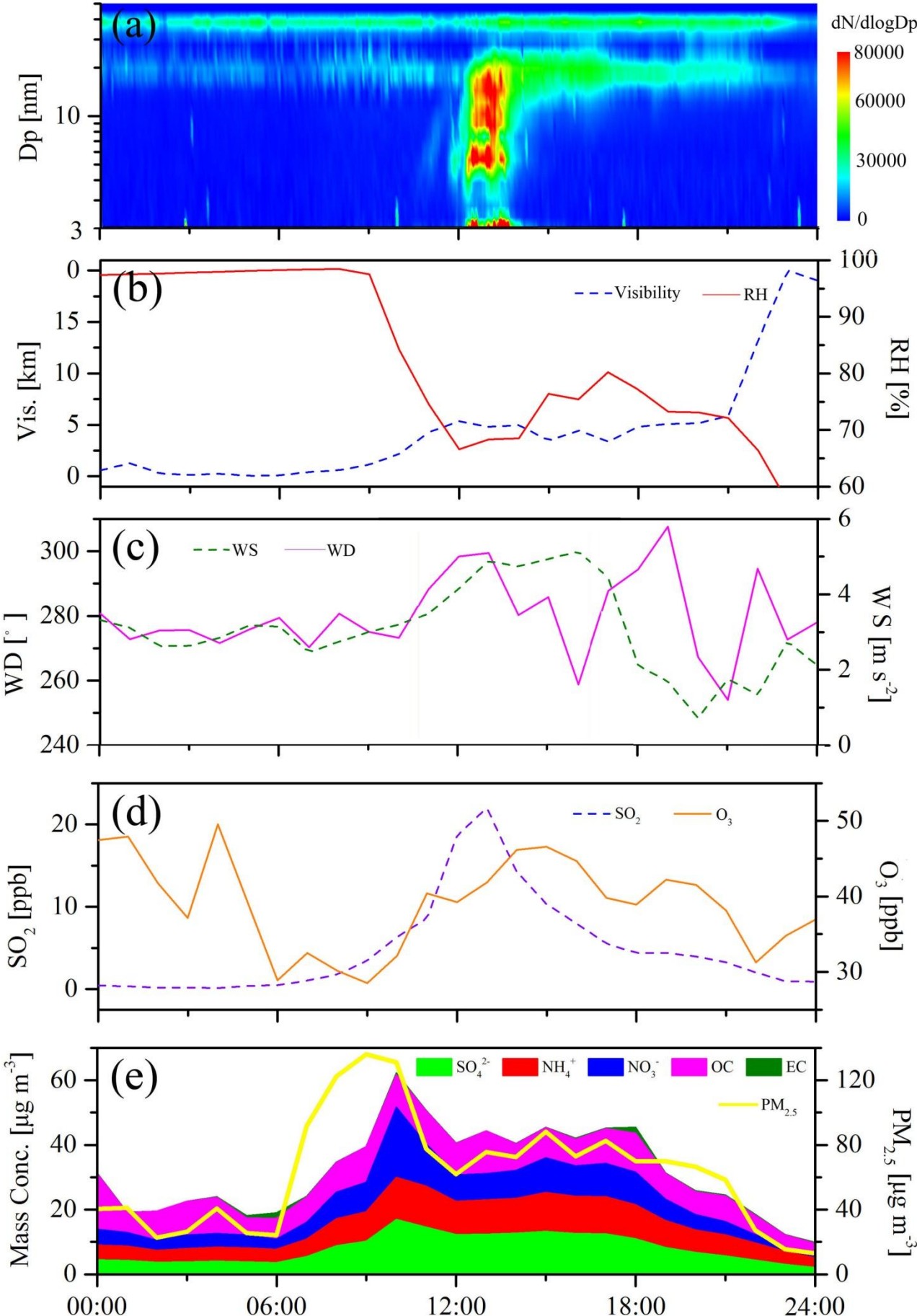

**Fig. 8.** Time series of particle size distribution, meteorological parameters, trace gases, chemical composition and mass concentration of $PM_{2.5}$ on 11 November 2014.

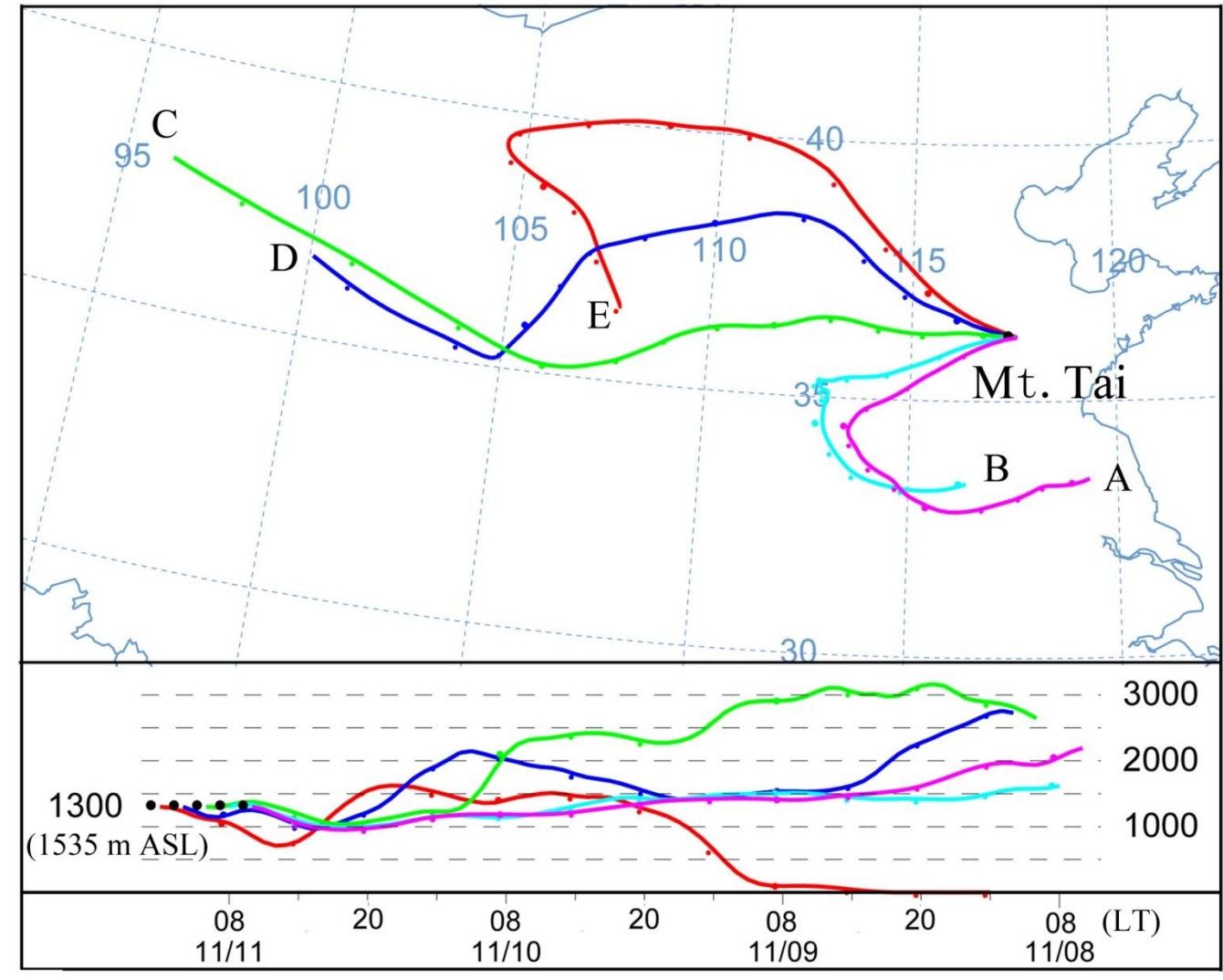

**Fig. 9.** Air mass backward trajectories for 72 h at 1535 m ASL at 6:00 (A), 8:00 (B), 10:00 (C), 12:00 (D) and 14:00 LT (E) on 11 November 2014.

**Table 1.** The calculated parameters of all the NPF events during the three campaigns, the minima and maxima are marked in blue and red, respectively

| Campaign | Date | $J_3$ cm$^{-3}$ s$^{-1}$ | $J_{3-20}$ cm$^{-3}$ s$^{-1}$ | GR nm h$^{-1}$ | CS $10^{-2}$ s$^{-1}$ | [H$_2$SO$_4$] $10^6$ cm$^{-3}$ | SO$_2$ ppb | O$_3$ ppb | Date | $J_3$ cm$^{-3}$ s$^{-1}$ | $J_{3-20}$ cm$^{-3}$ s$^{-1}$ | GR nm h$^{-1}$ | CS $10^{-2}$ s$^{-1}$ | [H$_2$SO$_4$] $10^6$ cm$^{-3}$ | SO$_2$ ppb | O$_3$ ppb |
|---|---|---|---|---|---|---|---|---|---|---|---|---|---|---|---|---|
| I | 26-Jul-14 | 3.34 | 13.32 | 1.54 | 0.5-3.1 | N/A | N/A | 56±7 | 8-Aug-14 | 2.61 | 16.30 | 1.15 | 1.8-15.9 | 2.87 | 2.7±2.3 | 65±10 |
|  | 27-Jul-14 | 0.94 | 6.12 | 1.55 | 0.6-2.1 | N/A | 2.4±2.0 | 56±9 | 11-Aug-1 | N/A | 1.44 | 1.71 | 0.2-6.9 | 3.81 | 0.6±0.5 | 56±6 |
|  | 2-Aug-14 | N/A | 3.60 | 3.81 | 0.1-8.7 | N/A | N/A | 22±13 | 12-Aug-1 | N/A | 40.13 | 3.69 | 0.8-7.4 | 22.1 | 10.8±7.9 | 79±16 |
|  | 3-Aug-14 | N/A | 1.69 | **7.76** | 0.1-26.9 | N/A | 1.5±1.2 | 35±17 | 15-Aug-1 | 2.75 | 4.52 | 3.00 | N/A | N/A | 1.0±1.0 | 63±12 |
|  | 6-Aug-14 | 23.90 | 52.54 | 5.78 | 0.2-1.9 | N/A | 4.8±2.6 | 50±5 | 20-Aug-1 | **0.82** | 1.33 | 4.80 | 0.6-2.7 | 9.25 | 5.2±4.3 | 72±8 |
|  | 7-Aug-14 | N/A | 10.90 | 1.45 | 0.1-24.8 | N/A | 1.5±1.7 | 56±4 |  |  |  |  |  |  |  |  |
| II | 22-Sep-14 | 0.99 | 1.58 | 1.12 | 0.7-9.0 | 7.74 | 5.7±1.7 | 76±8 | 4-Nov-14 | 7.73 | 15.06 | 0.77 | 0.6-1.5 | 5.39 | 6.3±1.2 | 56±3 |
|  | 29-Sep-14 | 16.63 | 54.97 | 1.13 | 0.7-21.1 | 2.43 | 1.4±1.1 | 48±5 | 6-Nov-14 | 0.98 | **1.10** | 1.45 | 0.1-1.2 | 2.02 | 0.6±0.5 | 13±5 |
|  | 30-Sep-14 | 7.94 | 20.61 | 1.62 | 0.5-28.4 | 1.09 | 1.9±0.5 | 2±1 | 7-Nov-14 | 1.40 | 5.59 | 1.26 | 0.1-2.0 | 0.93 | 0.4±0.3 | 23±11 |
|  | 2-Oct-14 | 5.16 | 12.02 | 1.15 | 0.2-6.6 | 1.98 | 1.6±1.7 | 41±4 | 8-Nov-14 | 9.40 | 13.53 | 2.95 | 0.4-2.3 | 6.71 | 4.1±2.6 | 38±3 |
|  | 3-Oct-14 | 5.44 | 14.07 | 1.50 | 0.1-4.6 | 1.34 | 0.7±0.7 | 42±3 | 10-Nov-1 | 1.02 | 2.17 | 3.41 | N/A | N/A | 2.3±0.7 | 60±4 |
|  | 5-Oct-14 | 8.13 | 26.41 | 2.00 | 0.5-6.9 | **0.52** | 0.2±0.1 | 11±5 | 11-Nov-1 | 3.29 | 14.42 | 2.52 | N/A | N/A | 7.8±8.3 | 34±5 |
|  | 6-Oct-14 | 2.54 | 8.94 | 0.78 | 0.4-3.4 | 0.53 | 0.3±0.4 | 7±2 | 12-Nov-1 | 12.39 | 26.36 | 1.09 | N/A | N/A | 1.6±0.4 | 35±2 |
|  | 8-Oct-14 | N/A | 1.40 | 0.92 | N/A | N/A | 5.0±0.6 | 71±6 | 14-Nov-1 | 9.92 | 13.41 | 1.55 | N/A | N/A | 6.1±4.0 | 40±2 |
|  | 10-Oct-14 | 6.54 | 27.04 | 1.80 | 0.2-1.6 | 4.33 | 1.8±2.1 | 54±3 | 16-Nov-1 | 5.36 | 11.06 | 2.47 | 0.5-1.2 | 11.6 | 5.7±2.9 | 34±6 |
|  | 11-Oct-14 | 9.43 | 20.37 | 0.70 | 0.2-2.2 | 1.50 | 3.1±2.5 | 68±6 | 17-Nov-1 | 9.15 | 16.82 | 1.10 | 0.2-0.9 | 8.38 | 1.5±0.2 | 38±2 |
|  | 13-Oct-14 | 6.17 | 26.84 | 1.64 | 0.4-2.0 | 7.43 | 5.1±1.6 | 27±8 | 18-Nov-1 | 20.45 | 57.11 | 0.80 | 0.1-1.6 | 4.42 | N/A | N/A |
|  | 14-Oct-14 | 10.15 | 18.11 | 1.55 | 0.1-1.4 | 1.43 | 0.9±0.8 | 44±4 | 19-Nov-1 | 8.89 | 15.41 | 2.51 | 0.3-2.8 | N/A | N/A | N/A |
|  | 15-Oct-14 | 6.76 | 30.54 | 2.76 | N/A | N/A | 10.1±2.9 | 60±6 | 22-Nov-1 | N/A | 11.60 | 1.23 | 0.3-5.6 | 4.18 | 5.3±4.9 | 10±8 |
|  | 16-Oct-14 | 6.14 | 36.96 | **0.58** | N/A | N/A | 1.5±1.9 | 41±1 | 24-Nov-1 | 11.03 | 24.51 | 1.50 | N/A | N/A | 2.9±2.1 | 26±10 |
|  | 17-Oct-14 | 1.55 | 5.75 | 2.12 | N/A | N/A | N/A | N/A | 25-Nov-1 | 7.11 | 10.97 | 2.02 | 0.1-2.9 | N/A | 0.5±0.4 | 25±13 |
|  | 18-Oct-14 | 6.16 | 22.89 | 2.03 | N/A | N/A | N/A | N/A | 1-Dec-14 | 4.96 | 7.55 | 1.77 | 0.1-0.9 | 1.22 | 1.4±0.3 | 27±4 |
|  | 21-Oct-14 | N/A | 1.98 | 2.36 | 0.4-1.3 | N/A | 0.3±0.3 | 13±5 | 3-Dec-14 | **25.04** | **57.43** | 1.51 | 0.4-1.4 | 1.64 | 12.9±9.6 | 13±11 |
|  | 24-Oct-14 | 3.64 | 8.99 | 1.25 | 0.5-1.3 | 1.12 | N/A | N/A | 4-Dec-14 | 4.80 | 10.62 | 2.69 | 0.4-1.4 | 5.48 | 2.9±0.3 | 13±9 |
|  | 25-Oct-14 | 7.50 | 14.74 | 1.60 | 0.4-1.4 | 0.89 | 0.3±0.3 | 15±11 | 5-Dec-14 | 3.34 | 5.83 | 1.32 | 0.2-1.2 | 0.95 | N/A | 4±1 |
|  | 27-Oct-14 | 7.24 | 15.49 | 0.99 | N/A | N/A | 0.2±0.3 | 12±4 | 6-Dec-14 | N/A | 1.52 | 1.99 | N/A | N/A | 2.2±1.2 | 4±9 |
|  | 28-Oct-14 | 2.98 | 13.17 | 1.21 | N/A | N/A | 0.2±0.1 | 5±2 | 7-Dec-14 | N/A | 11.64 | 1.42 | N/A | N/A | 7.9±2.5 | 8±5 |
|  | 2-Nov-14 | 11.24 | 17.78 | 0.72 | 0.2-0.9 | 4.97 | 0.7±0.1 | 33±3 | 8-Dec-14 | 9.78 | 16.33 | 1.24 | N/A | N/A | 2.4±1.4 | 16±10 |
|  | 3-Nov-14 | 8.77 | 18.57 | 1.10 | 0.3-1.3 | 6.65 | 4.6±0.6 | 37±3 |  |  |  |  |  |  |  |  |
| III | 16-Jun-15 | 1.45 | 9.22 | 3.98 | 1.7-4.9 | 13.6 | 6.3±0.8 | 105±7 | 4-Jul-15 | 12.55 | 23.25 | 0.91 | N/A | N/A | 5.7±2.9 | 95±20 |
|  | 20-Jun-15 | 6.16 | 32.50 | 3.28 | 0.5-1.8 | 10.5 | 2.8±1.6 | 81±11 | 8-Jul-15 | N/A | 8.25 | 0.93 | 0.6-2.3 | N/A | 0.5±0.5 | 90±16 |
|  | 21-Jun-15 | 3.90 | 6.08 | 3.44 | 1.4-3.0 | **25.7** | 10.0±2.1 | 105±15 | 13-Jul-15 | N/A | 5.64 | 1.95 | 1.0-2.4 | 1.10 | 0.1±0.1 | 110±15 |
|  | 2-Jul-15 | 9.61 | 43.41 | 1.08 | 0.2-1.9 | N/A | 2.2±1.3 | 69±10 | 15-Jul-15 | N/A | 19.72 | 2.86 | 0.7-1.6 | N/A | 0.2±0.1 | 88±12 |
|  | 3-Jul-15 | 5.54 | 10.86 | 2.65 | 0.2-2.1 | N/A | 5.5±2.5 | 101±13 | 25-Jul-15 | N/A | 19.56 | 1.88 | 0.4-1.9 | 2.41 | 0.3±0.2 | 99±9 |

**Table 2.** Summary of averages, medians, 25th percentiles, 75th percentiles, minima and maxima for the calculated parameters on the basis of Table 1

| | Average | Minimum | Maximum | 25th percentile | Median | 75th percentile |
|---|---|---|---|---|---|---|
| $J_3$ $(cm^{-3}\ s^{-1})$ | 7.10 | 0.82 | 25.04 | 3.31 | 6.15 | 9.41 |
| $J_{3\text{-}20}$ $(cm^{-3}\ s^{-1})$ | 16.61 | 1.10 | 57.43 | 6.12 | 13.47 | 20.61 |
| GR $(nm\ h^{-1})$ | 1.98 | 0.58 | 7.76 | 1.15 | 1.55 | 2.51 |
| CS $(10^{-2}\ s^{-1})$ | 1.4 | 0.1 | 28.4 | 0.5 | 0.9 | 1.7 |
| $[H_2SO_4]$ $(10^6\ cm^{-3})$ | 5.23 | 0.52 | 25.7 | 1.28 | 3.34 | 7.07 |
| $SO_2$ (ppb) | 3.2 | 0.1 | 12.9 | 0.7 | 2.2 | 2.7 |
| $O_3$ (ppb) | 45 | 2 | 110 | 19 | 41 | 66 |

**Table 3.** Comparison of NPF characteristics between Mt. Tai and other studies in China

| Observation site | FR ($cm^{-3} s^{-1}$) | GR ($nm\ h^{-1}$) | Freq. | Data | Air mass style | Ref. |
|---|---|---|---|---|---|---|
| Mt. Tai | $7.10\pm5.39$ ($J_3$) | $1.98\pm1.27$ ($GR_{3-20}$) | 40 % | Jul-Dec 2014 & Jun-Aug 2015 | Mountain (1534 m ASL) | This study |
| Mt. Tai Mo Shan | 0.97-10.2 ($J_{5.5}$) | 1.5-8.4 ($GR_{5.5-25}$) | 33 % | Oct-Nov 2010 | Mountain (640 m ASL ) | Guo et al. (2012) |
| Mt. Huang | 0.09-0.30 ($J_{10}$) | 1.42-4.53 ($GR_{10-20}$) | 37 % | Apr-Jul 2008 | Mountain (1840 m ASL ) | Zhang et al. (2016) |
| Mt. Huang | | 2.29-4.27 ($GR_{10-15}$) | 18 % | Sep-Oct 2012 | Mountain (869 m ASL ) | Hao et al. (2015) |
| Mt. Daban | | 0.8-3.2 | 79 % | Sep-Oct 2013 | Mountain (3295 m ASL ) | Du et al. (2015) |
| SouthYellow Sea & East China Sea | 0.3-15.2 ($J_{5.6-30}$) | 2.5-5.0 | 16 % | Oct-Nov 2011 & Nov 2012 | Marine | Liu et al. (2014) |
| Backgarden | 2.4-4.0 ($J_{3-25}$) | 4.0-22.7 ($GR_{3-25}$) | 25 % | Jul 2006 | Rural | Yue et al. (2013) |
| Nanjing | 2.6 ($J_6$) | 10.4 ($GR_{6-30}$) | 44 % | Dec 2011-Nov 2013 | Suburban | Qi et al. (2015) |
| Lanzhou | | 1.2-16.9 ($GR_{10-20}$) | 33 % | Jun-Jul 2006 | Suburban | Gao et al. (2012) |
| Xinken | 0.5-5.2 ($J_{3-20}$) | 2.2-19.8 ($GR_{3-20}$) | 26 % | Oct-Nov 2004 | Suburban | Liu et al. (2008) |
| Shanghai | 2.3-19.2 ($J_3$) | 1.9-38.3 ($GR_{7-20}$) | 21 % | Nov 2013-Jan 2014 | Urban | Xiao et al. (2015) |
| Nanjing | 1.6-6.7 ($J_{10-25}$) | 5.6-9.6 ($GR_{10-25}$) | 40 % | Jul-Aug 2012 | Urban | An et al. (2015) |
| Beijing | 5.0-44.9 ($J_3$) | 1.86-6.7 ($GR_{7-30}$) | 26 % | Jul-Sep 2008 | Urban | Wang et al. (2015) |
| Lanzhou | 0.2-6.2 ($J_{14.6-25}$) | 2.6-12.3 ($GR_{14.6-25}$) | 34 % | Aug-Nov 2014 | Urban | Zhang et al. (2017) |
| Qingdao | 13.3 ($J_{5.6-30}$) | 2.0-10.2 | 41 % | Apr-May 2010 | Urban | Zhu et al. (2014) |
| Hong Kong | 1.9 ($J_{5.5}$) | 3.7-8.3 ($GR_{5.5-10}$) | 23 % | Dec 2010-Jan 2011 | Urban | Wang et al. (2014a) |

**Table 4.** The parameters on four NPF haze days and the averages of all the NPF days during 6:00-18:00 LT

| | T (°C) | RH (%) | WS (m s⁻¹) | SO₂ (ppb) | O₃ (ppb) | NO₂ (ppb) | CO (ppb) | PM₂.₅ (μg m⁻³) |
|---|---|---|---|---|---|---|---|---|
| Average | 5.4 | 54 | 2.1 | 3.9 | 31 | 10.5 | 759 | 31 |
| 26-Jul-14 | 19.6±2.1 | 67±10 | 0.7±0.4 | 5.4±3.4 | 65±12 | N/A | N/A | 42±3 |
| 17-Oct-14 | 12.8±2.6 | 59±11 | 2.7±0.4 | N/A | N/A | 10.0±4.2 | 827±256 | 121±97 |
| 18-Oct-14 | 14.0±2.0 | 55±8 | 1.5±0.4 | N/A | N/A | 7.1±1.9 | 757±40 | 49±18 |
| 11-Nov-14 | 4.8±1.9 | 82±12 | 3.7±1.1 | 8.1±6.7 | 38±6 | 11.9±3.1 | 1641±284 | 84±30 |