# Peer review of "Investigation of new particle formation at the summit of Mt. Tai, China"

_Atmospheric Chemistry and Physics, 2016_

## Referee Comment (RC1) · Anonymous Referee #2 · 6 Nov 2016

This study focused on the new particle formation at a mountain site in Easter China. The manuscript for sure provides very large and valuable dataset referring to new particle formation at a mountain site in China. However, the current manuscript is not well organized and there is few in-deep thoughts on the determinants and mechanism of the NPF events at this site. Besides, some of the conclusions drawn in the mu manuscript are questionable. Therefore, major revisions are needed before publication should be considered.

Specific comments:

1. The language of this manuscript is far from a scientific publication. It is strong recommended that the manuscript should be carefully revised, probably by a native English speaker. There are many grammatical mistakes, incorrect omissions and collocations,
even in the abstract, including but not limited to the following items.

(1) Page 1, line 17, “appeared” should be “appears”

(2) Page 2, line 3, there is a “that” missing after “exhibited”.

(3) Page 2, line 6, “Recent decades” should be “In recent decades”.

(4) Page 2, line 17, should “after then” be “after that”?

(5) Page 3, line 10, does the author want to say “What’s the contribution of estimated gaseous sulfuric acid on nucleation and growth processes?”

(6) Page 5, line 17, “in Table 3 it also exhibited the characteristics comparison between Mt. Tai and some other typical recent researches in China.” should be “In table 3, it is also exhibited that the characteristics. . .”

(7) Page 6, line 1. “the sampling season”.

(8) Page 6, line 15. “Although gaseous sulfuric acid formed through sulfur dioxide photochemical reactions was involved in NPF, neither high sulfur dioxide being strong NPF nor low value limiting NPF burst.” This sentence is too vague to be understood.

(9) Page 7, line 18. “Figure 3 picked 40 days continuous data from. . ..” can be revised as “Figure 3 exhibits the continuous data from . . .”.

(10) Page 7, line 21, “exhibited” should be “exhibits”. “This result was in accordance with. . .” should be “This result is in accordance with. . .”

(11) Page 7, line 23, better to substitute “improve” with “increase”.

(12) line 29, “Existent of ozone could quantify the oxidation capacity and photochemical activities in the atmosphere, directly reacting with related species such as VOCs and indirectly affecting sulfuric acid formation via hydroxyl radical production”. Incorrect adverbial use.

(13) Page 8, line 1. Should be “it is found that . . .”. “revealed” should be “experienced”.

(14) Page 8, line 3. What does “made for” mean here?

(15) Page 8, line 6. “Day-to-day analysis revealed that” should be “reveal”

(16) Page 8, line 10. “it meant” can be revised as “This suggests’

(17) line 12, “Particle number concentrations depending on wind direction in each mode were not obvious, and none of directions always showed significantly higher or smaller particle concentrations and it had clear difference with reports in Nanjing.” This sentence is too vague to be understood.

(18) line 14, should be “this suggests that few direct. . .”

(19) line 18, “illustrated” should be “illustrates”.

(20) Page 9, line 2. Revised as “Fig. 6 illustrates the data from. . .”

(21) Page 10, line 5. “In contrast” instead of “by contrast”.

(22) line 11, what does “background total nucleation particles” mean?

(23) line 30. “It revealed that . . .” What does “it” here stand for?

2. Page 1, line 25, “PM2.5 variation was always in accordance with particle total volume concentration.” This is nearly common sense. I don’t understand why this can be a conclusion in this paper.

3. Page 3, line 10. I am not sure whether the third point the author made here can be a real “scientific question”. The author may need to contribute some in-deep thoughts here.

4. Page 4. The author need to define all the parameters used in the equations.

5. Page 4, line 15. Why the Fgrowth can be neglected in this study? Does the author have evidence on this?

6. Page 4, line 23. The paper written by Mikkonen et al. has provided a more precise

H2SO4 estimation equation, in which another two parameters, CS and RH were used. Why don't the author use this one?

7. Page 6, line 10. "NPF events could be observed in each month, and frequent NPF occurrence was in campaign ĐŸ which showed the frequency of 56 % (others were only 21 % by contrast). It could be interpreted that campaign ĐĘ and ĐÍ were in rainy and foggy season, and such wet condition seemed adverse to NPF." The campaign ĐĘ, ĐŸ and ĐÍ should be defined before.

8. Page 5, line 24. "Another notable period was from 10 October to 18 October 2014, during which it had frequent NPF events and most of formation rates were larger than 75th percentile (20.61 cm-3 s-1). It could be associated with specific atmospheric conditions because of sudden temperature drop." Why the sudden temperature drop increased the frequency of NPF?

9. Page 6, line 1. "Reasons for our large value were possibly not only related to geographically wide mountaintop location, but also sampling season impacts and size range difference for calculating formation rate." What does the "geographically wide mountaintop location" mean? What specifically are the "season impacts" and "size range difference"?

10. Page 6, line 20. Why it didn't exhibit significant distinction in the condensation sink values between NPF and non-NPF days? The author needs to provide a clear explanation.

11. Page 6, line 30. The uncertainty is not only from the solar radiation data, but also from the calculation itself as all the parameters for the equations are estimated based on the data in EU, not in China.

12. Page 8, line 1. "it found ozone concentration revealed slight drop during nucleation process on many NPF days and ozone consumption reactions might take place." What does the author mean here? The ozone concentration decreased because of the

reactions during nucleation process? Why is that?

13. Page 8, line 3. “our statistical results showed that NPF preferred to occur on clear or partial cloudy daytime.” What is the statistical evidence here?

14. Page 8, line 6. “Day-to-day analysis revealed that temperature and relative humidity always had cyclic variation, and NPF events preferably occurred on high temperature and low relative humidity conditions. High temperature and low humidity could promote vertical transportation and photochemical reactions in the atmosphere.” I don’t think this is something that can be “revealed”. NPF always occurs in the middle of the day when temperature is higher and humidity is lower, because it is driven by photochemistry. This phenomenon cannot be logically concluded as high temperature and low relative humidity favour the NPF.

15. Page 8, line 15. “It could suggest that few direct particle pollution sources existed around observation site and nucleation might be the primary source for particles on Mt. Tai.” I don’t understand how the author made this conclusion. If there is no local sources nearby, particles can be from transport from other regions.

16. Page 8, line 24. “Compared with air masses coming from cleaner western parts of China, air masses going through Beijing et al. polluted areas had more complicated components and enhanced NPF events.” Is there any evidence in this paper to say so?

17. Page 8, line 29. “Hence NPF events with local continental backward trajectories were more vulnerable to local point sources.” This is confusing. The author just discussed that there is few direct sources around the observation site. How does the author make this conclusion then?

18. Page 9, line 9. Why “the higher atmospheric humidity in campaign ĐĘ might enhance the sticking possibility of particles for molecules”? Does the author refer to liquid phase or the phase change due to water content?

19. Page 9, line 17. “But particles recombination in close sizes could contribute to

the growth after nucleation and higher PM2.5 within limiting values possibly increased this possibility." Coagulation of particles within the nucleation mode is always negligible under ambient condition. The author needs to provide strong evidences to prove that it is important in this study.

20. Page 10, line 10. The author try to calculate the particle density on clean days here. In this case, the author needs to prove that the PM2.5 measurement was very accurate at low concentration level. Also, the particle growth factor should be considered if there is no dryer before the PM2.5 monitor.

21. Page 10, line 17. "NPF had small impact on total volume concentration on polluted day, which might be related to large background fine particles." I don't know what the author wants to express here.

22. Page 10, line 23, is there a definition of "haze day" before? If it means PM25 > 75 ug m-3, why the average was only 64 ug m-3?

23. Page 11, line 8. "Reason for the former was possibly clusters recombination, and the latter decrease might be on account of energy threshold (i.e., nucleation barrier) and 10 atmospheric scavenging." Again, the author need to provide very strong prove that the coagulation of clusters is very important in this study.

---

## Referee Comment (RC2) · Anonymous Referee #1 · 9 Nov 2016

General comments: This study investigates the new particle formation at a high mountain site of China. Since there were limited studies on NPF at mountain site of China, this work can provide useful data about NPF at high altitude area of China. However, this manuscript is not well organized, and lack of necessary in-deep analysis. A lot of conclusions were made arbitrarily. Some statements were not persuasive. I suggest that the manuscript should be major revised before published in ACP.

Specific comments:

1. The language requires polished. I highly suggest that the authors ask a native speaker to edit this manuscript. Some sentences in this manuscript are too obscure to understand.

2. P1, Line 14, campaign I and II seem overlapped. Please check the campaign period.

3. P1, Line 18, Mt. Tai CANNOT show larger formation rate. Besides "larger formation rate" is misleading. this work only studied less than one year NPF, it's difficult to conclude the formation rate here is large. Considering the gas concentration, the FR for clean site should be lower.

4. P1, Line 22, what does "limited higher PM 2.5" mean here? Usually higher particle contribution inhabits the new particle formation. Anyway, I guess the authors were trying to say that during the relatively polluted days, the GR is higher? It's because of the gas concentration, but has nothing to do with PM concentration.

5. P1, Line 20, what does "proxy" mean?

6. P1, Line 22, recombination? Do author want to say "coagulation"?

7. P1, Line 23, "haze" is inaccurate.

8. P2, Line 7, change "around" to "over", what does "refer to" mean?

9. Session 2.1, more information about the site should be provided, e.g. the height of the site or inlet from ground.

10. P3 Line 25, "Its measurement range of. . ." should be rewritten.

11. Session 2.3.2, the sulfuric acid estimation method used here already has very large uncertainty. Besides, the accuracy of the radiation from HYSPLIT model is far from enough for sulfuric acid estimation.

12. Equation 6, the literatures here are old. For the current knowledge, sulfuric acid is considered to contribute to the nucleation, but negligibly to the particle growth. The authors should rethink about the discussion about this.

13. P5, Line 10, change "each" to "every"

14. Session 3.1, authors should analyze what controls the occurrence of NPF, source or sink?

15. Fig. 1, y should be SO2*OH to present NPF source. Comparing SO2 with CS makes no sense.

16. Fig. 3, change the color bar so that one can see the "banana curve" clearly.

17. Session 3.2, because the sulfuric acid estimation has large uncertain, author should reconsider how to analyze this part.

18. P7, Line 22, this interpretation has no evidence. It's more from author's guess, not from data. There are a few interpretations like this, e.g. line 25.

19. Session 3.4, PM2.5 is not directly related to NPF. Again, it should be discussed that it is source or sink that control the occurrence of NPF.

20. Session 3.5, how to combine the NAIS and WPS data. Especially these two instruments have overlapped size range. Maybe this should be included in the experimental session

21. Session 3.6, the definition of "haze" here is unclear. Visibility is not a crucial criterion for haze. The PM concentrations are similar for so called haze and non-haze days. Can it be haze or fog? Also it seems redundant and reduplicated to session 3.5.

---

## Author Comment (AC1) · 19 Dec 2016

**Investigation of new particle formation at the summit of Mt. Tai, China**

**Ganglin Lv et al.**

We are very grateful to the detailed comments offered by the Referee #1. We have revised the manuscript accordingly, and listed below in red are our point-by-point responses.

Response to Referee #1

1. The language requires polished. I highly suggest that the authors ask a native speaker to edit this manuscript. Some sentences in this manuscript are too obscure to understand.
Answer: The manuscript has been revised by a native English speaker. The sentences have been marked in red.

2. P1, Line 14, campaign I and II seem overlapped. Please check the campaign period.
Answer: We have checked the campaign period, and there was a spelling mistake in campaign I. The right period of campaign I was from 25 July to 24 August 2014, and we have corrected it.

3. P1, Line 18, Mt. Tai CANNOT show larger formation rate. Besides "larger formation rate" is misleading. this work only studied less than one year NPF, it's difficult to conclude the formation rate here is large. Considering the gas concentration, the FR for clean site should be lower.
Answer: Thanks. We have rewritten the related discussion in page 6, line 9-19. The observed NPF events at the summit of Mt. Tai were indeed very strong in our study, with the $J_3$ of $7.10\pm5.39$ cm$^{-3}$ s$^{-1}$. The deep-analysis for its mechanisms might need the precise measurement for precursors, and we couldn't make the accurate explanation for it because of limited measurement instruments. However, intensive precursor transport in region (eastern China) and enhanced photochemical activities at the summit of Mt. Tai could, at least partly, explain the higher formation rate values.

4. P1, Line 22, what does "limited higher PM 2.5" mean here? Usually higher particle contribution inhabits the new particle formation. Anyway, I guess the authors were trying to say that during the relatively polluted days, the GR is higher? It's because of the gas concentration, but has nothing to do with PM concentration.
Answer: We appreciated the better understanding you made for "limited higher PM $_{2.5}$", and we actually wanted to express the higher growth rate under the relatively polluted environment. However, the discussion of PM$_{2.5}$ and particle growth has been removed in the revised manuscript.

5. P1, Line 20, what does "proxy" mean?
Answer: In the manuscript, "proxy" means "proximity for sulfuric acid". In our study, direct measurement for gas-phase sulfuric acid was unavailable. Instead, the proxy for sulfuric acid was estimated based on solar radiation, sulfur dioxide, condensation sink and relative humidity.

6. P1, Line 22, recombination? Do author want to say "coagulation"?
Answer: Thanks. We actually wanted to show the "self-coagulation". However, the related sentences have been removed in the revised manuscript.

7. P1, Line 23, "haze" is inaccurate.

Answer: Thanks. We have replaced "haze" to "hazy"

8. P2, Line 7, change "around" to "over", what does "refer to" mean?

Answer: We have corrected "around" to "over", and we have also changed the "refer to" to "include".

9. Session 2.1, more information about the site should be provided, e.g. the height of the site or inlet from ground.

Answer: We have added a more detailed description about the site, seen in page 3, line 17-26. The observation site is almost the peak of Mt. Tai, so the height of the site is near 1534 m ASL. All the instruments were installed inside a large container, sampling through short inlet tubes from the container at a height of about 3 m above the ground level.

10. P3 Line 25, "Its measurement range of…" should be rewritten.

Answer: Thanks. We have rewritten the related sentences in Sect. 2.2, please see page 4, line 3-5.

11. Session 2.3.2, the sulfuric acid estimation method used here already has very large uncertainty. Besides, the accuracy of the radiation from HYSPLIT model is far from enough for sulfuric acid estimation.

Answer: In our revised manuscript, we have used the non-liner type proxies for sulfuric acid estimation in page 5, line 8-14, and both CS and RH have been taken into consideration in the new estimation method. In this study, direct measurement for solar radiation was not possessed, and the meteorological station in China did not have the related data as well. There were no studies which reported the approximate values at the adjacent locations. As the result, the HYSPLIT model is the only available for solar radiation at the summit of Mt. Tai. We made the comparisons between the reported solar radiation and estimated solar radiation from HYSPLIT in some other locations. Results showed the absolute value of solar radiation from HYSPLIT might involve some error with the real solar radiation in the atmosphere, but its diurnal variation pattern could be believable to a certain extent. Except for the calculated sulfuric acid proxy concentrations in page 7, line 24-31 and Table 1, most of results in the revised manuscript were based on the comparison between NPF days and non-NPF days. The comparison method could weaken the effect of absolute solar radiation on the results. In addition, although the calculated sulfuric acid proxy concentrations have some error, its magnitude ($\sim10^6$ orders of magnitude for initial sulfuric acid concentrations at the summit of Mt. Tai, $cm^{-3}$) should be believable.

12. Equation 6, the literatures here are old. For the current knowledge, sulfuric acid is considered to contribute to the nucleation, but negligibly to the particle growth. The authors should rethink about the discussion about this.

Answer: Many recent researches, such as Meng et al. (2014), showed that sulfuric acid made a minor contribution on particle growth and the oxidation production of VOCs accounted for the major contribution. In consideration of the inaccurate estimation of absolute sulfuric acid values, the discussion of sulfuric acid contribution on the NPF has been removed in the revised manuscript.

13. P5, Line 10, change "each" to "every"

Answer: We have corrected it.

14. Session 3.1, authors should analyze what controls the occurrence of NPF, source or sink?

Answer: Thanks very much. Multiple factors affecting the occurrence of NPF have been added in Sect. 3.2, page 6-11, including condensation sink, sulfuric acid proxy, sulfur dioxide, ozone, temperature, relative humidity, wind direction and air mass transport. In Sect. 3.2.1, we focus on the effects of sulfuric acid source and condensation sink, seen in page 7, line 1-31 and page 8, line 1-21.

15. Fig. 1, y should be SO2*OH to present NPF source. Comparing SO2 with CS makes no sense.

Answer: Thanks. SO2*OH may not be possessed in this study, but we use the sulfur dioxide proxy concentration as the Y axis in Fig. 2 to compare the source and sink on NPF days and non-NPF days.

16. Fig. 3, change the color bar so that one can see the "banana curve" clearly.

Answer: We have changed the color bar and used the ORIGIN 9.0 to make the contour plot in our revised manuscript.

17. Session 3.2, because the sulfuric acid estimation has large uncertain, author should reconsider how to analyze this part.

Answer: Thanks. As the 11$^{th}$ answer, we focus on the comparisons between NPF days and non-days in the revised manuscript, and avoid involving absolute values as possible as we can. Detailed discussion was seen in Set. 3.2.1, page 7, line 24-31 and page 8, line 1-30.

18. P7, Line 22, this interpretation has no evidence. It's more from author's guess, not from data. There are a few interpretations like this, e.g. line 25.

Answer: There might be confused expression in our manuscript, and new discussion for sulfur dioxide was seen in page 9, line 1-16. Because photochemical reactions of $SO_2$ are the major source for sulfuric acid at the summit of Mt. Tai (direct sulfuric acid could be neglected) and NPF was indeed favorable to the high $SO_2$ concentration, it is plausible that higher $SO_2$ concentration can increase the possibility of rich precursors for NPF.

19. Session 3.4, PM2.5 is not directly related to NPF. Again, it should be discussed that it is source or sink that control the occurrence of NPF.

Answer: In the revised manuscript, the discussion for $PM_{2.5}$ variation has been removed. Instead, we focused on the NPF event during hazy episodes, and one NPF event on 11 November 2014 was analyzed in detail to explore the factors affecting its occurrence. Please see in Sect. 3.3, page 11, line 15-31 and page 12, line 1-17.

20. Session 3.5, how to combine the NAIS and WPS data. Especially these two instruments have overlapped size range. Maybe this should be included in the experimental session.

Answer: Thanks very much. Combination of the NIAS and WPS is a very good suggestion. It was a pity that the particle number size distributions measured by two instruments matched not very well in this study, and the WPS data were discontinuous because of frequent instrument maintaining. Therefore, we mainly used the NAIS data in the analysis of the NPF events, and the WPS data were the assist. However, we can adopt the combination of the NAIS and WPS in our future study.

21. Session 3.6, the definition of "haze" here is unclear. Visibility is not a crucial criterion for haze. The PM concentrations are similar for so called haze and non-haze days. Can it be haze or fog? Also it seems redundant and reduplicated to session 3.5.

Answer: In the revised manuscript, the hazy episode can be identified when the atmospheric visibility is less than 10 km and the RH is less than 80 % simultaneously, seen in page 11, line 17-19. The fog episode is identified when the atmospheric visibility is less than 10 km and the RH is more than 90 %. The crucial criterion between haze and fog is the RH. In the revised manuscript, we focus on the factors affecting the occurrence of NPF, and the related discussion for particle behaviors has been removed.

---

## Author Comment (AC2) · 19 Dec 2016

**Investigation of new particle formation at the summit of Mt. Tai, China**

Ganglin Lv et al.

We are very grateful to the detailed comments offered by the Referee #2. We have revised the manuscript accordingly, and listed below in red are our point-by-point responses.

Response to Referee #2

1. The language of this manuscript is far from a scientific publication. It is strong recommended that the manuscript should be carefully revised, probably by a native English speaker. There are many grammatical mistakes, incorrect omissions and collocations, even in the abstract, including but not limited to the following items.
Answer: Thanks. The manuscript has been revised by the native English speaker. The sentences have been marked in red.

(1)  Page 1, line 17, "appeared" should be "appears"
Answer: We have corrected the word. However, as we revised the abstract, the part which includes this word has been removed.

(2) Page 2, line 3, there is a "that" missing after "exhibited".
Answer: We have corrected it.

(3) Page 2, line 6, "Recent decades" should be "In recent decades".
Answer: Thanks, we have corrected it.

(4) Page 2, line 17, should "after then" be "after that"?
Answer: We used "thereafter" to replace "after then".

(5) Page 3, line 10, does the author want to say "What's the contribution of estimated gaseous sulfuric acid on nucleation and growth processes?"
Answer: Yes, we appreciated the better expression you offered and we have corrected this sentence. However, the discussion for contribution of sulfuric acid has been removed in the revised manuscript, so this sentence has also been removed. Thanks very much.

(6) Page 5, line 17, "in Table 3 it also exhibited the characteristics comparison between Mt. Tai and some other typical recent researches in China." should be "In table 3, it is also exhibited that the characteristics…"
Answer: We have corrected the sentence.

(7) Page 6, line 1. "the sampling season".
Answer: We have revised the word as the "the observation period". However, the related sentence has been removed in the revised manuscript.

(8) Page 6, line 15. "Although gaseous sulfuric acid formed through sulfur dioxide photochemical reactions was involved in NPF, neither high sulfur dioxide being strong NPF nor low value limiting NPF burst." This sentence is too vague to be understood.

Answer: We wanted to show that the sulfur dioxide concentration didn't directly affect the occurrence of NPF, although sulfuric acid could be formed through the photochemical reactions of sulfur dioxide. In the revised manuscript, the related sentence has been removed.

(9) Page 7, line 18. "Figure 3 picked 40 days continuous data from..." can be revised as "Figure 3 exhibits the continuous data from…".

Answer: We appreciated the better expression you made. However, the related sentence has been removed in the revised manuscript.

(10) Page 7, line 21, "exhibited" should be "exhibits". "This result was in accordance with…" should be "This result is in accordance with…"

Answer: Thanks, and we have corrected them. However, the related part has been removed in the revised manuscript.

(11) Page 7, line 23, better to substitute "improve" with "increase".

Answer: Thanks, and we have corrected it.

(12) line 29, "Existent of ozone could quantify the oxidation capacity and photochemical activities in the atmosphere, directly reacting with related species such as VOCs and indirectly affecting sulfuric acid formation via hydroxyl radical production". Incorrect adverbial use.

Answer: We have revised this sentence as "$O_3$ has been considered to quantify the oxidation capacity and photochemical activities in the atmosphere, directly reacting with related species such as VOCs and indirectly affecting sulfuric acid formation via hydroxyl and hydroperoxy radicals" in the revised manuscript.

(13) Page 8, line 1. Should be "it is found that…". "revealed" should be "experienced".

Answer: We have corrected them. However, the related sentence has been removed in the revised manuscript.

(14) Page 8, line 3. What does "made for" mean here?

Answer: The "made for" means "be in favor of". However, we have revised the sentence as "favorable meteorological conditions could promote the occurrence of the NPF when precursors were insufficient in the atmosphere" in the revised manuscript.

(15) Page 8, line 6. "Day-to-day analysis revealed that" should be "reveal"

Answer: Thanks, and we have corrected it. However, the related sentence has been removed in the revised manuscript.

(16) Page 8, line 10. "it meant" can be revised as "This suggests'

Answer: Yes, we have corrected it.

(17) line 12, "Particle number concentrations depending on wind direction in each mode were not obvious, and none of directions always showed significantly higher or smaller particle concentrations and it had clear difference with reports in Nanjing." This sentence is too vague to be understood.

Answer: New discussion is in the revised manuscript, please see in page 10, line 22-26. In the revised manuscript, we found that the accumulation mode particles showed a high level of concentration when the corresponding wind was in the northeast, which could partly explain the less occurrence of NPF from the northeast wind direction. In addition, we found that the nucleation and Aitken modes particles were approximate possibility in each wind direction if there was not the NPF event, suggesting that few direct particle sources for nucleation and Aitken modes particles existed around the observation site.

(18) line 14, should be "this suggests that few direct…"

Answer: We appreciated the better expression you made. But we have rewritten the related sentences, please see in page 10, line 22-26.

(19) line 18, "illustrated" should be "illustrates".

Answer: Thanks, and we have corrected it.

(20) Page 9, line 2. Revised as "Fig. 6 illustrates the data from…"

Answer: We have corrected it. However, the related sentence has been removed in the revised manuscript.

(21) Page 10, line 5. "In contrast" instead of "by contrast".

Answer: We have corrected it. However, the related sentence has been removed in the revised manuscript.

(22) line 11, what does "background total nucleation particles" mean?

Answer: The "background total nucleation particles" means "the particle number concentration of 3-20 nm without the effect of NPF events or other air mass transport". In the revised manuscript, the related sentence has been removed.

(23) line 30. "It revealed that…" What does "it" here stand for?

Answer: "it" stands for "the day on 11 November 2014" in the paper. However, the related discussion has been removed in the revised manuscript.

2. Page 1, line 25, "$PM_{2.5}$ variation was always in accordance with particle total volume concentration." This is nearly common sense. I don't understand why this can be a conclusion in this paper.

Answer: We have deleted it in the revised manuscript.

3. Page 3, line 10. I am not sure whether the third point the author made here can be a real "scientific question". The author may need to contribute some in-deep thoughts here.

Answer: Thanks. The discussion related to particle behaviors has been removed in the revised manuscript.

4. Page 4. The authors need to define all the parameters used in the equations.

Answer: We have made the modification in the revised manuscript.

5. Page 4, line 15. Why the $F_{growth}$ can be neglected in this study? Does the author have evidence on this?

Answer: We have added a brief explanation in page 4, line 25-26. Based on particle size distribution data and duration of nucleation process, we first estimated the range of growth rate and found that particles rarely grow beyond 20 nm before formation ended. In our study, most newly formed particles grew slowly during the NPF events. In some cases of NPF events, particles grow faster but its duration was fairly short. As a result, the $F_{growth}$ could be neglected in this paper. In addition, many reported papers such as Dal Maso (2005), have also neglected the growth loss.

6. Page 4, line 23. The paper written by Mikkonen et al. has provided a more precise H2SO4 estimation equation, in which another two parameters, CS and RH were used. Why don't the author use this one?

Answer: Thanks very much. In our revised manuscript, we have used the non-liner type proxies for sulfuric acid in Sect. 2.3.2, and both CS and RH have been taken into consideration.

7. Page 6, line 10. "NPF events could be observed in each month, and frequent NPF occurrence was in campaign ÐŸ which showed the frequency of 56 % (others were only 21 % by contrast). It could be interpreted that campaign Ð ¿ E and Ð´l were in rainy and foggy season, and such wet condition seemed adverse to NPF." The campaign Ð ¿ E, ÐŸ and Ð´l should be defined before.

Answer: We have added them in the introduction (page 3, line 11) and discussion (page 5, line 21-22).

8. Page 5, line 24. "Another notable period was from 10 October to 18 October 2014, during which it had frequent NPF events and most of formation rates were larger than 75th percentile (20.61 cm-3 s-1). It could be associated with specific atmospheric conditions because of sudden temperature drop." Why the sudden temperature drop increased the frequency of NPF?

Answer: Sudden temperature drop might increase the coal or biomass burning in the region, which possibly increased the emission for precursors. The elevated precursor concentration was in favor to the occurrence of NPF. However, we consider that this phenomenon is lack of accurate evidence, so the related discussion has been removed in the revised manuscript.

9. Page 6, line 1. "Reasons for our large value were possibly not only related to geographically wide mountaintop location, but also sampling season impacts and size range difference for calculating formation rate." What does the "geographically wide mountaintop location" mean? What specifically are the "season impacts" and "size range difference"?

Answer: The "geographically wide mountaintop location" means "there is not the obvious obstacle around the mountain-top observation site". The wide landform may be in favor of precursor transport from the other locations. The observation site in our study is the summit of Mt. Tai without obstacles nearby, where air mass transport may be intensive. The "geographically wide mountaintop location" may not be an appropriate use, and the new discussion is in page 6, line 17-19.

The "season impacts" means "influence of the different season", and the "size range difference" means "the different particle interval for calculating the formation rate, such as the $J_3$ and $J_5$". Because they may not be the special characteristics of Mt. Tai, we have removed the results in the revised manuscript.

10. Page 6, line 20. Why it didn't exhibit significant distinction in the condensation sink values between NPF and non-NPF days? The author needs to provide a clear explanation.

Answer: We have added a more detailed analysis of CS on NPF days and non-NPF days in Set. 3.2.1, please see in page 7, line 12-23 and page 8, line 6-9.

11. Page 6, line 30. The uncertainty is not only from the solar radiation data, but also from the calculation itself as all the parameters for the equations are estimated based on the data in EU, not in China.

Answer: We have made a new discussion related to the sulfuric acid in Set. 3.2.1, please see in page 7, line 24-31 and page 8, line 1-5. In the revised manuscript, we compare the estimated sulfuric acid proxy concentration with reports in China rather than in EU.

12. Page 8, line 1. "it found ozone concentration revealed slight drop during nucleation process on many NPF days and ozone consumption reactions might take place." What does the author mean here? The ozone concentration decreased because of the reactions during nucleation process? Why is that?

Answer: There may be confused expression in our manuscript, and we have made a more detailed and accurate discussion of ozone in Set. 3.2.2, seen in page 9, line 17-30. The diurnal variation of ozone concentration on NPF days showed a slight drop near the sunset. During this period, the residual ozone, with the high concentration at the summit of Mt. Tai, might involve in sulfuric acid formation via producing the hydroxyl and hydroperoxy radicals under the solar radiation condition. The decreased ozone concentration is not caused by the nucleation, but because of the production of the hydroxyl and hydroperoxy radicals or ozone photolysis.

13. Page 8, line 3. "our statistical results showed that NPF preferred to occur on clear or partial cloudy daytime." What is the statistical evidence here?

Answer: In this study, approximately 90 % of all the NPF events occurred on the clear or partial cloudy daytime, seen in page 10, line 2-3.

14. Page 8, line 6. "Day-to-day analysis revealed that temperature and relative humidity always had cyclic variation, and NPF events preferably occurred on high temperature and low relative humidity conditions. High temperature and low humidity could promote vertical transportation and photochemical reactions in the atmosphere." I don't think this is something that can be "revealed". NPF always occurs in the middle of the day when temperature is higher and humidity is lower, because it is driven by photochemistry. This phenomenon cannot be logically concluded as high temperature and low relative humidity favour the NPF.

Answer: Our previous understanding for temperature and relative humidity might be one-sided, and we have made a new detailed discussion of temperature and relative humidity in Set. 3.2.3, seen in page 10, line 3-17. In the revised manuscript, we compare the average diurnal variations of the temperature and relative humidity during NPF days and non-NPF days. The results showed that the lower temperature and relative humidity conditions seemed to be favorable for the occurrence of the NPF.

15. Page 8, line 15. "It could suggest that few direct particle pollution sources existed around observation site and nucleation might be the primary source for particles on Mt. Tai." I don't

understand how the author made this conclusion. If there is no local source nearby, particles can be from transport from other regions.

Answer: There might be confused expression in our manuscript, and we have rewritten this part in 10, line 22-26. In this study, we mainly want to discuss the particle number concentration distribution as the function of wind direction. During all the observation days, the accumulation mode particles showed the higher values when the wind came from northeast. So we speculate that this exceptional phenomenon may partly contribute to the less NPF occurrence in the northeast wind direction. In addition, we found that the nucleation and Aitken modes particle in all wind directions were almost evenly distributed if there was not NPF events, and none of directions showed significantly higher or smaller values. However, there were uneven and similar distributions between nucleation and Aitken modes particles during all the observation days. These phenomena suggest that NPF events possibly result in above difference. In addition, wind directions just reflect a local situation rather than the long-distance transport. If there were local point sources around observation sites, some directions should be obviously higher particle (nucleation and Aitken modes) concentrations. So we speculate that there are few direct local sources for nucleation and Aitken modes particles around the site.

16. Page 8, line 24. "Compared with air masses coming from cleaner western parts of China, air masses going through Beijing et al. polluted areas had more complicated components and enhanced NPF events." Is there any evidence in this paper to say so?

Answer: We have revised the related part, and the new detailed description could be seen in page 11, line 1-14. During all the NPF days, continental air masses accounted for 80 % of the total air masses, among which four-fifths passed though polluted areas (Beijing, Hebei Province, Shanxi Province, Henan Province, Shaanxi Province) before reaching the observation site. In contrast, 63 % of the total air masses were continental air masses during all the non-NPF days, among which only two-fifths passed though polluted areas before reaching the observation site. Most of the continental air masses on non-NPF days came from the south (cleaner part of China) or transported over Bohai Sea and Yellow Sea. From the statistical results, it seems plausible that air masses going through the polluted areas could increase the occurrence of NPF. When air masses passed though the polluted areas, it might bring precursors or motivating substances for NPF.

17. Page 8, line 29. "Hence NPF events with local continental backward trajectories were more vulnerable to local point sources." This is confusing. The author just discussed that there is few direct sources around the observation site. How does the author make this conclusion then?

Answer: There might be confused expression in our manuscript. The afore-mentioned conclusion was based on the discussion of particle number concentration distribution with wind directions, and there were few local point sources for nucleation and Aitken modes particles around the observation site. Air mass backward trajectories were based on the long-time air mass transport. In this study, all the local air masses during NPF days were upward, which might bring local pollutants to the sites to affect the NPF. However, we have removed this conclusion in the revised manuscript because a few local air mass back trajectories were observed in this study.

18. Page 9, line 9. Why "the higher atmospheric humidity in campaign ĐE¿ might enhance the sticking possibility of particles for molecules"? Does the author refer to liquid phase or the phase change due to water content?

Answer: Many literatures, such as Mikkonen et al (2011), reported that high RH might increase the sticking probability of molecules to existing particles. In our study, we didn't consider the liquid phase or the phase change. However, the discussion for $PM_{2.5}$ variation has been removed in the revised manuscript,

19. Page 9, line 17. "But particles recombination in close sizes could contribute to the growth after nucleation and higher PM2.5 within limiting values possibly increased this possibility." Coagulation of particles within the nucleation mode is always negligible under ambient condition. The author needs to provide strong evidences to prove that it is important in this study.

Answer: In the revised manuscript, the discussion related to $PM_{2.5}$ and particle growth has been removed.

20. Page 10, line 10. The author try to calculate the particle density on clean days here. In this case, the author needs to prove that the PM2.5 measurement was very accurate at low concentration level. Also, the particle growth factor should be considered if there is no dryer before the PM2.5 monitor.

Answer: Thanks very much. You have given me a very important instruction for calculating the particle density, and we think our instrument may not meet requirement for very accurate measurement at low concentration level and we will try to seek other method for calculating the particle density. However, the related discussion for particle behaviors under clean and polluted conditions has been removed in the revised manuscript.

21. Page 10, line 17. "NPF had small impact on total volume concentration on polluted day, which might be related to large background fine particles." I don't know what the author wants to express here.

Answer: On the polluted days with elevated $PM_{2.5}$ concentration, the fine particles may account for the major volume of the total particle volume concentration. However, the fine particle concentration changed a little before and after NPF events on the polluted days, suggesting that the NPF events made the minor contribution to the fine particles and the total particle volume concentration. In the revised manuscript, the related discussion has been removed.

22. Page 10, line 23, is there a definition of "haze day" before? If it means PM25 > 75 ug m-3, why the average was only 64 ug m-3?

Answer: In our revised manuscript, we use an accurate "hazy episodes" to replace "haze day". Because the occurrence of NPF was only observed in the daytime, so the "hazy episodes" might be fitter in our study. The hazy episode can be identified when the atmospheric visibility is less than 10 km and the RH is less than 80 % simultaneously, seen in page 11, line 17-19. The value of 64 ug m$^{-3}$ just represented the average value at the summit of Mt. Tai. In fact, the average $PM_{2.5}$ concentration on hazy days/episodes at the foot of mountain is much larger than 64 ug m$^{-3}$.

23. Page 11, line 8. "Reason for the former was possibly clusters recombination, and the latter decrease might be on account of energy threshold (i.e., nucleation barrier) and atmospheric scavenging." Again, the author need to provide very strong prove that the coagulation of clusters is very important in this study.

Answer: Thanks. In the revised manuscript, the discussion related to particle behaviors has been

removed.

---

## Author Comment (AC7) · 19 Dec 2016

The comment was uploaded in the form of a supplement:
http://www.atmos-chem-phys-discuss.net/acp-2016-806/acp-2016-806-AC7-supplement.pdf

---

## Author Response (AR2)

**Investigation of new particle formation at the summit of Mt. Tai, China**

Ganglin Lv et al.

We are very grateful to the detailed comments offered by the Referee #1 and Referee #2. We have revised the manuscript

5    accordingly, and listed below in red are our point-by-point responses.

**Response to Referee #1**

1. This version is much better than the previous one. The topic is worth publishing, and content is scientifically sound.

10    However, it's still not satisfactory. The English writing still requires polishing. Some words used in the text are strange and

not professional. I highly suggest to asking a native speaker to edit the MS, or ask for professional English writing service.

Besides the MS is not well organized. I only list one example:P7 Line 7 This paragraph is not well organized. The authors

claimed their results are "similar" to Wang et al.'s , but they did not show any results, which makes the reader very confused,

although they showed the results afterwards, the writing is too ramble to have people understand.

15    Answer: Thanks. The manuscript has been revised by the native English speaker. Besides, this adjusted version has been

reorganized, and such example in P7 Line 7 has also been avoided. The related modifications have been marked in red in

manuscript.

**Response to Referee #2**

1. The authors discussed in the manuscript about the influence of O3 concentration on NPF frequency. But the O3 data shown in figure 4 seems strange. Why the O3 concentration kept stable at a high level before Nov. 17th, 2014? Why there is no diurnal variation these days? Is the O3 data trustable? Besides, it is better to provide PM2.5 data in Figure 4.

Answer: Thanks very much. The data in Fig. 4 are trustable. $O_3$ concentration has the obvious seasonal variation, and its background concentration at the summit of Mt. Tai is relatively high. In Fig. 1d, it showed the general (average) diurnal variation of $O_3$ concentration at the summit of Mt. Tai. Generally, the diurnal variation of $O_3$ concentration at the summit of Mt. Tai had two prominent features, a trough in the morning (because of dry deposition) and a peak in the afternoon (because of $O_3$ formation by solar radiation), and these features were in good agreement with results in Sun et al. (2016). However, day-to-day $O_3$ concentration is also influenced by the air mass transport, meteorological conditions et al., so sometimes $O_3$ concentration may not absolutely behave like Fig. 1d. Before Nov. 17th, 2014 in Fig. 4, the high level of $O_3$ concentrations were mainly contributed by the high values of $O_3$ precursors (such as $NO_2$) in early November. During these days, $O_3$ concentration generally had the minor diurnal variations (especially for trough and peak features), detailed in FigureR1 below.

[Figure]

FigureR1 $O_3$ concentration during 11 November to 17 November 2014

In our study, we observed that the hourly average $O_3$ concentrations on NPF days were always lower than that on non-NPF days, indicating that lower $O_3$ environment may benefit NPF events at the summit of Mt. Tai. $O_3$ could enhance the atmospheric oxidation capacity, but it could bring some negative pollutants simultaneously. During NPF events at the summit of Mt. Tai, the positive influence of the increase in $O_3$ may not always offset the negative influence of accompanying pollutants. In our study, it did not find what numerical $O_3$ concentration would be in favor of NPF, and merely concluded that such environment with the relatively lower $O_3$ concentration would be beneficial for NPF at the summit of Mt. Tai. The detailed description is shown in page 9, line 29-31 and page 10, line 1-11.

The $PM_{2.5}$ and visibility have been added in Fig. 4d.

2. In part 3.2.3, the authors argue that "the gas point sources from the above two directions can increase the probability of the NPF occurrence in a certain extent". Could the authors provide some more evidence, e.g., SO2 concentration, to prove this statement?

Answer: Thanks. As shown in manuscript, the NPF events preferred to occur in the east-southeast (85 °-110 °) and west-southwest (250 °-300 °) wind directions at the summit of Mt. Tai during our observations. The phenomena could be explained from the sinks and sources. The CS in the east-southeast wind direction showed relatively low value of $1.5 \times 10^{-2}$ s$^{-1}$, which was much less than the average CS at the wind directions between 40 ° and 110 ° ($2.0 \times 10^{-2}$ s$^{-1}$). In the west-southwest wind direction, it corresponded to the elevated $SO_2$ concentration of 4.3 ppb, which was much larger than the other wind directions (such as the average $SO_2$ concentrations of 2.0 ppb at the wind directions between 220 ° and 350 °). The more detailed description could be seen in page 11, line 4-15.

3. Also in part 3.2.3, the authors attribute the high frequency of NPF in Mongolia air mass to "transported air masses brought high level of anthropogenic precursors from the polluted areas". Is there clear evidence on this, e.g., higher $SO_2$ and $PM_{2.5}$? Or could it due to the low aerosol loading caused by the long range air masses?

Answer: Thanks. We have made the related discussion in the adjusted manuscript, seen in page 12, line 4-19. In the manuscript, continental air mass was the majority of all the backward trajectories, which could be classified into two categories based on transported regions: the polluted continental air mass (Type I) and the clean continental air mass (Type II). Type I passed though the heavily polluted areas of Beijing, Hebei Province, Shanxi Province, Henan Province, and Shaanxi Province, whereas Type II was either from the south China or transported over the Bohai Sea (B-S) and Yellow Sea (Y-S). In this study, the average $SO_2$ concentrations in Type I and Type II during NPF days were 3.9 ppb and 1.2 ppb, respectively. A prominent increase of average $SO_2$ concentration was in Type I, suggesting that enhanced precursors could be carried to Mt. Tai when the air masses passed over the heavily polluted areas. The average mass concentrations of $PM_{2.5}$ in Type I and Type II were 33 µg m$^{-3}$ and 23 µg m$^{-3}$, and Type I had higher value of $PM_{2.5}$. In addition, the average chemical composition of $PM_{2.5}$ between Type I and Type II showed significant difference. Type I had elevated percentages of sulfate, ammonium and nitrate, which indicated that transported air masses passed through polluted S-rich and N-rich areas which might provide potential species for NPF.

4. Page 5, line 23, "and the frequent occurrence…". Is there a "highest" before the" frequent occurrence"?

Answer: The related sentences have been rewritten in the revised manuscript, please see in page 6, line 3-5.

5. Page 8, line 11, should "might be explained that…" be "might be explained by …"?

Answer: Thanks. We have replaced "that" to "by".

**References**

[revised manuscript text omitted]

---

## Author Response (AR3)

**Investigation of new particle formation at the summit of Mt. Tai, China**

Ganglin Lv et al.

We are very grateful to the detailed comments offered by the Referee #1 and Referee #2. We have revised the manuscript accordingly, and listed below in red are our point-by-point responses.

**Response to Referee #1**

1. the author still hasn't proven the accuracy of O3 data. There is no diurnal variation of O3 in the first few days, but very strong variation in the following days. On contrary, the PM2.5 and SO2 concentrations varied greatly in the first few days. The author discussed about the high NO2 concentration in the first few days, which is not provided. Is the NO2 also as stable as O3?

Answer: Thanks, the data of $O_3$ in this study are accurate and trustable. During our observations, there were also some similar periods during which the concentration of $O_3$ kept stable and high level, such as period from 12 October to 16 October 2004.

In Figure R1, it showed the $NO_2$ concentration during 11 November to 16 November 2014. As like concentrations of $PM_{2.5}$ and $SO_2$, $NO_2$ concentration also varied frequently in this period. $NO_2$ was indeed an important precursor for $O_3$, but it was not the only one. Several factors, such as stratosphere-troposphere exchange, biomass burning, meteorological conditions and precipitation, could also affect the $O_3$ concentration in the atmosphere, which might contribute the result during 11 November to 17 November 2014.

[Figure]

Figure R1 $NO_2$ concentration during 11 November to 16 November 2014

2. Page 2, line 24,another paper which summarized the NPF at diversified sites in China is recommended here.

Peng, J. F., Hu, M., Wang, Z. B., Huang, X. F., Kumar, P., Wu, Z. J., Guo, S., Yue, D. L., Shang, D. J., Zheng, Z., and He, L. Y.: Submicron aerosols at thirteen diversified sites in China: size distribution, new particle formation and corresponding contribution to cloud condensation nuclei production, Atmos. Chem. Phys., 14, 10249-10265, doi:10.5194/acp-14-10249-2014, 2014.

Answer: Thanks, we have cited it in the Introduction in Page 2.

3. It is not quite convictive when the author discussed "factors affecting the occurrence of NPF" only by comparing the averaging values of each factor at NPF and non-NPF days. For example, on page 10, the author attribute the low O3 concentration on NPF days to the "negative accompanying pollutants". However, such expression is strange. As far as I know, on one ever found a "negative influence" of O3 on NPF before. The author could just explain this as co-linear with some other pollutants, instead of the "negative influence". Also, some

Answer: Thanks. We have adjusted the manuscript, and further discussions have been added. In addition, the "negative influence" has been replaced by "co-linear", seen in Page 10, line 8-11.

4. Fig.1, it is better to provide error bars for each data point.

Answer: we have changed as the error bars in new Fig. 1.

5. Fig.7, it is not very clear what the SO2 bars stand for here unless spend some time to find out. It is recommended to the author to change some other way to express the SO2 concentration.

Answer: Thanks very much, and we have modified this Figure, seen in Page 26.

**Response to Referee #2**

1. The introduction is insufficient. Nucleation and growth not only has great impact on climate (CCN here in the manuscript, also radiation forcing Kazil et al., ACP, 2010), but also air quality (i.e. haze, Guo et al. PNAS 2014), and human health (. Nucleation mechanism, the roles of different precursors, sulfuric acid, ammonium, amine, organics (recent Nature papers). Also, there were several mountain site NPF studies in the world. The authors should review these studies, not only the ones in China.

Answer: Thanks very much. The introduction has been revised, and more papers and researches have been cited. The details could be seen in revised manuscript.

2. My biggest concern about the measurement is that can NAIS and WPS data well combine together. Two instruments have overlap data bins. How's the data? Do they agree well? If not, how to combine them? As far as I know, these two techniques usually don't agree very well.

Answer: Yes, the NIAS and WPS really had the overlapping data bins, and data measured by two instruments did not agree very well in our study. In the manuscript, we mainly focused on the nucleation particles, and data measured by the NAIS instrument were more trustworthy. Therefore, we mainly used the NAIS data in the analysis of the NPF events, and the WPS data were just the assist. In addition, the WPS data were discontinuous because of frequent instrument maintaining in the study. Combination of the NIAS and WPS is a very good suggestion, and we could consider it in the future study.

3. The estimation of Sulfuric acid proxy. The method itself already has large uncertainty. Moreover, the radiation used here is from HYSPLIT model. HYSPLIT model cannot accurately model the radiation for single site. This caused even larger uncertainty.

Answer: Thanks, there are really errors in the estimation of sulfuric acid proxy through the HYSPLIT model. In this study, direct measurement for solar radiation was not possessed, and the meteorological station in China did not have the related data as well. There were not any study which reported the approximate values at the adjacent locations. As the result, the HYSPLIT model is the only available for solar radiation at the summit of Mt. Tai. We made the comparisons between the reported solar radiation and estimated solar radiation from HYSPLIT in some other locations. Results showed the absolute value of solar radiation from HYSPLIT might involve some error with the real solar radiation in the atmosphere, but its diurnal variation pattern could be believable to a certain extent. In addition, although the calculated sulfuric acid proxy concentrations have some error, its order of magnitude ($\sim 10^6$ orders of magnitude for initial sulfuric acid concentrations at the summit of Mt. Tai, $cm^{-3}$) should be believable.

In the adjusted manuscript, we have cut most of discussions related to sulfuric acid proxy, except for the calculated sulfuric acid proxy concentrations in page 9, line 2-10 and data in Table 1 for getting a rough estimation of gaseous sulfuric acid concentration at the summit of Mt. Tai .

4. 3.2.1 and 3.2.2 seem talk about the similar things.

Answer: We have merged them together in the adjusted manuscript.

5. Fig.2 I can't see any relationship during NPF days.

Answer: In the adjusted manuscript, the former Fig.2 and its related discussions have been removed.

6. Please explain why NPF prefer low temperature. I cannot figure out the mechanism here. Correlation doesn't mean relationship

Answer: Thanks. We have adjusted the manuscript, and the expression and words are used more cautiously. Based on the current techniques, we couldn't explain the actual mechanism between NPF and lower temperature in the study, but there have been some reports in which it discussed the potential mechanism for them. For example, Guo et al. (2012) reported that the bind between sulfuric acid and water molecules was easier in lower temperature.

7. The NPF has nothing to do with wind speed and direction. It's no use to analyze this. The real reason for this is the relationship between source (sulfuric acid) and sink (CS).

Answer: Yes, we have removed the discussion of wind speed in the adjusted manuscript. Because wind directions reflect the local situation for air mass, we merged the analysis of wind direction into the Section 3.2 of sinks and sources. The details can be seen in Page 9, line 21-31.

8. Fig. 6 I cannot see any difference for NPF and non-NPF.

Answer: Fig. 6 shows the air mass back trajectories on NPF days and non-NPF days. Based on the transport range and distance, all the air mass backward trajectories were classified into three categories of continental air mass, local air mass and maritime air mass. The differences between NPF days and non-NPF days were mainly :(1) the ratios of three categories; (2) Type I and Type II of the continental air masses; (3) the percentage and transport routes of maritime air mass. The detailed discussions for this Figure could be seen in Page 11, line 19-31 and Page 13, line 1-10.

9. The "haze" here is odd. I don't think this session makes aby sense.

Answer: Thanks. We have adjusted the manuscript, and related discussions about haze have been removed.

[revised manuscript text omitted]

---

## Author Response (AR4)

**Investigation of new particle formation at the summit of Mt. Tai, China**

Ganglin Lv et al.

We are very grateful to the detailed comments offered by the Referee #1, Referee #2 and Co-Editor. We have revised the manuscript, and listed below in red are our responses.

**Response to Referee #1**

I think this version is fine now. Only one thing, the English still needs to be polished. After that, I think this MS can be published.

Answer: Thanks for the positive report. The English of the manuscript has been polished, and the sentences have been marked in red.

**Response to Referee #2**

No advice.

**Response to Co-Editor**

My overall feeling with your manuscript is that it provides little physical insight into the issue of NPF from your field measurements. For example, you concluded that the O3 concentration was lower during NPF events, but indicated that the continental air masses passing through more polluted areas may contribute to an increased frequency of NPF events. Such statements are self-contradicting. Also, you concluded that the continental air masses passing through more polluted areas (denoted as Type I) could enhance the occurrence of NPF, which was possibly associated with high level of precursors carried from the polluted regions. However, no discussion was provided for any species that likely contributed to such enhanced NPF. For example, it has been shown that sulfuric acid likely plays a key role in NPF in China (The roles of sulfuric acid in new particle formation and growth in the mega-city of Beijing, Atmos. Chem. Phys. 10, 4953, 2010). Also, organic species can also enhance NPF (Atmospheric new particle formation enhanced by organic acids, Science 304, 1487, 2004). Those aspects need to be assessed in your manuscript.

Also, the statement "Field observations of NPF at the mountain-top sites are scarce in China, and the results of such studies could significantly contribute to atmospheric aerosol pollution control" was unsubstantiated on the basis of your research.

In addition, one of the reviewers still suggested to improve the writing of your manuscript, which I agree with. I provide a few examples below for the English issues.

In the abstract, change "in a period" to "during a period". Also, please fix the phrase "were mainly through affecting". Add

an article "the" before "maritime air mass". Change "seemed not to be favorable" to "did not correspond to a favorable".

P. 13, line 27, please fix the phrase "an increased the frequency of NPF events".

Answer: Thanks very much. We have revised the discussion of $O_3$ issue, seen in Page 10, line 6-23. Elevated $O_3$ concentration is beneficial to NPF, which has been identified by many previous researches. The lower $O_3$ concentration on NPF days in our study might be associated with high level of $NO_x$ concentration on Mt. Tai. Previous studies showed that there was an $NO_x$ turnaround value of 10-15 ppb for the $O_3$ formation, and $O_3$ production would decrease as $NO_x$ when $NO_x$ concentration exceeded the turnaround value. In our study, the $NO_x$ concentration was at a high level, probably leading to the decreased $O_3$ concentration on NPF days as we observed.

We have revised the discussion related to the continental air mass of Type I, seen in Page 12, line 7-22. We compared the average chemical composition of $PM_{2.5}$ and $SO_2$ concentration in Type I and Type II on NPF days. Results showed that a prominent increase of $SO_2$ concentration was found in Type I, suggesting that sulfuric acid played an important role in NPF for Type I. In addition, the significantly elevated mixing ratios of ammonium and nitrate were found in Type I, suggesting that potential ammonia also promoted NPF. Besides Type I air mass, we also added some discussion of precursor species in other parts, such as the Sect. 3.5 in adjusted manuscript. These papers you recommended and other related published papers have been cited in adjusted manuscript.

The sentence "Field observations of NPF at the mountain-top sites are scarce in China, and the results of such studies could significantly contribute to atmospheric aerosol pollution control" has been changed as "For the NPF issue, although a few observational results have been reported in China, to date very few researches conducted the comprehensive analysis of favorable conditions that gave rise to NPF on mountain-top sites, not to mention investigating characteristics of NPF events based on the improved instruments.". Seen in Page 13, line 29-30 and Page 14, line 1.

The writing of this manuscript has been revised, and the sentences have been marked in red. These English issues you listed have been modified in adjusted manuscript.

[revised manuscript text omitted]